# InternAgent-DR: Advancing deep research with dynamic structured knowledge flow

## Abstract

Deep research is an inherently challenging task that demands both breadth and depth of thinking. It involves navigating diverse knowledge spaces and reasoning over complex, multi-step dependencies, which presents substantial challenges for agentic systems. To address this, we propose InternAgent-DR (Deep Research), a multi-agent framework that actively constructs and evolves a dynamic structured knowledge flow to drive subtask execution and reasoning. InternAgent-DR is capable of strategically planning and expanding the knowledge flow to enable parallel exploration and hierarchical task decomposition, while also adjusting the knowledge flow in real time based on feedback from intermediate reasoning outcomes and insights. InternAgent-DR achieves state-of-the-art performance on both general and scientific benchmarks, including GAIA, HLE, GPQA and TRQA, demonstrating its effectiveness in multi-disciplinary research scenarios and its potential to advance scientific discovery. The code is available at `https://github.com/Alpha-Innovator/InternAgent`.

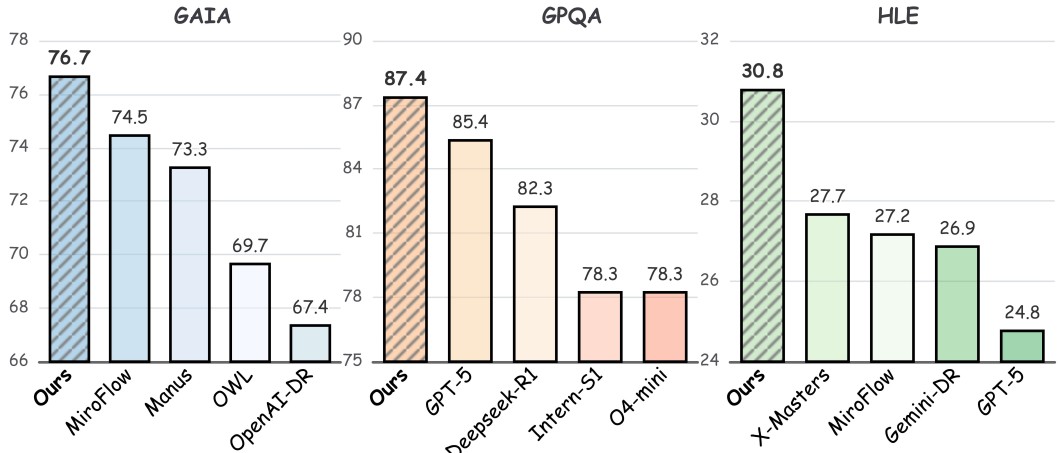

Figure 1: InternAgent-DR (Ours) achieves leading performance on the GAIA, GPQA, and HLE benchmarks, outperforming competitive agent workflow methods (OpenAI-DeepResearch, MiroFlow, Manus, OWL) as well as LLM-based approaches (GPT-5, Intern-S1, DeepSeek-R1).

## 1 Introduction

The rapid progress of Large Language Models (LLMs) (Guo et al., 2025; OpenAI, 2025a; Yang et al., 2025; Bai et al., 2025) has marked a significant milestone in artificial intelligence, exhibiting impressive capabilities in natural language understanding, generation, and complex reasoning (Yao et al., 2023b; Trivedi et al., 2024). Researchers have found that the capabilities of large language models (LLMs) can be harnessed to build agent systems for handling diverse tasks (Schick et al., 2023; Fan et al., 2024). More notably, LLMs are proving to be valuable tools in facilitating scientific research and discovery (Team et al., 2025; Zhang et al., 2025b). However, effectively leveraging these capabilities in open-ended research contexts remains a non-trivial challenge—demanding not

only iterative hypothesis formulation and strategic information acquisition, but also the orchestration of multi-step reasoning within dynamic and often uncertain knowledge spaces. These considerations have motivated the development of Deep Research (DR) systems—frameworks that transcend isolated reasoning or passive retrieval by integrating LLMs within structured, goal-directed workflows. The design of such systems is essential for unlocking the full potential of LLMs in enabling systematic scientific discovery across various domains.

Existing deep research systems (Hu et al., 2025; Team, 2025) often draw inspiration from either individual or collaborative research paradigms. **(1) Single-agent paradigm:** (Wu et al., 2025; Tao et al., 2025; Li et al., 2025b; Yao et al., 2023b) LLMs centrally manage the research workflow by leveraging a long context window to accumulate and reason over information. While this setup mirrors the behavior of an individual researcher, it is prone to tunnel vision—overcommitting to early hypotheses and lacking the breadth necessary for wide-ranging exploration. **(2) Multi-agent paradigm:** (Hu et al., 2025; man, 2025; Team et al., 2025) such designs scale up research efforts by leveraging explicit planning and role specialization among agents. However, the prevalent reliance on serial plan execution in these systems places strict demands on context management. Excessively retrieved information may result in context window overflow and easily overwhelm the system (Huang et al.), thereby reducing the system's ability to sustain deep and coherent reasoning. Overall, both paradigms face inherent trade-offs between exploratory breadth and reasoning depth, highlighting the need for more adaptive, reflective, and context-sensitive research agents.

In this work, we present InternAgent-DR, a multi-agent system built upon a dynamic knowledge flow framework that enables structured and efficient knowledge propagation throughout the process of scientific discovery. The system begins with a flow planner that constructs an initial knowledge flow, where nodes represent subproblems to be solved or key concepts to be retrieved, and edges encode the knowledge dependencies among them. As discovery progresses, the knowledge flow can be incrementally expanded to ensure both exploratory breadth and reasoning depth, while its structure remains dynamically revisable in response to intermediate findings—enabling the overall process to proceed in a reflective and adaptive manner. Importantly, each node in the flow not only guides the execution of subtasks but also supports recursive decomposition, integration of upstream knowledge, and local summarization of intermediate results. This leads to a more refined and contextually relevant knowledge stream, which allows for deep reasoning within local regions of the flow while maintaining global coherence through dynamic flow-level adjustments. Such a design ensures the efficiency of the knowledge flow and enhances the system's ability to perform complex, multi-step scientific problem solving.

To demonstrate the superior performance of InternAgent-DR, we conduct experiments on several challenging benchmarks, including GAIA (Mialon et al., 2023), which evaluates the general problem-solving abilities of AI assistants, as well as three scientific-question-answering benchmarks HLE (Phan et al., 2025),GPQA (Rein et al., 2024) and TRQA (Zhang et al., 2025b). InternAgent-DR achieves state-of-the-art performance on GAIA, HLE and TRQA, and demonstrates highly competitive results on GPQA. These findings highlight the strong problem-solving capability of InternAgent-DR, enabled by the integration of graph-driven planning. In summary, our main contribution can be described as follows:

- We propose a novel dynamic structured knowledge flow to encode the logic in complex problem solving, enabling the deep research agent to explicitly capture dependencies among subproblems and key concepts, in contrast to conventional frameworks.

- We develop InternAgent-DR, a novel multi-agent system for deep research built upon the dynamic structured knowledge flow, capable of generating structured plans and dynamically refining them during execution to enhance performance.

- We evaluate InternAgent-DR on the general AI assistant benchmark GAIA and the multi-disciplinary scientific question-answering benchmarks HLE, GPQA and TRQA, demonstrating state-of-the-art performance across all of them.

## 2 INTERNAGENT-DR

InternAgent-DR is built upon a dynamic knowledge flow that enables structured and adaptive scientific research. As illustrated in Figure 2, the system comprises three core components: **Knowledge**

**Flow Planner**, which constructs high-quality knowledge flows tailored to the research objective; **Knowledge Collector**, which executes subtasks and enriches each node with relevant contextual information; and **Knowledge Flow Refiner**, which monitors progress and dynamically adjusts the flow based on intermediate outcomes and newly acquired knowledge. By enabling multiple agents to collaborate along this evolving flow, InternAgent-DR achieves systematic, scalable, and efficient problem solving. We begin by formalizing the concept of the structured knowledge flow, followed by detailed descriptions of Knowledge Flow Planner, Knowledge Collector, and Knowledge Flow Refiner.

## 2.1 STRUCTURED KNOWLEDGE FLOW

Structured Knowledge Flow provides principled guidance for systematically organizing information to improve both the systematicity and effectiveness of deep research.

A common workflow in deep research agents is to address a user query $q$ by assembling a strictly linear pipeline $L(q) = [s_1, s_2, \ldots, s_n]$ and executing it in order $s_1 \to s_2 \to \cdots \to s_n$. The precedence relations are implicitly encoded by positional order, i.e., $s_i \prec s_j \iff i < j$. Despite its procedural simplicity and ease of implementation, such a linear formalism fails to capture the inherently complex and non-linear dependencies of real-world research processes.

To better capture the complex structure of deep research reasoning processes, we adopt a directed acyclic graph $G = (V, E)$ to explicitly model both task dependencies and knowledge flow. Each node $v_i \in V$ is a typed subtask node $v_i = (t_i, d_i, s_i, c_i)$, where $t_i \in \{search, solve, answer\}$ is the task type, $d_i$ is the task desciption, $s_i$ is the execution state of the node and $c_i$ is the resulting knwoledge context of the node if successfully executed. Each directed edge $e_{ij} = (v_i, v_j, r_{ij}) \in E$ specifies how the output of $v_i$ conditions or constrains $v_j$ using the relation $r_{ij} \in R$, where $R$ is the set of relation types. This flow makes precedence among the nodes and supports parallel execution on independent branches, yielding a more expressive and verifiable substrate for Deep Research.

As an illustration, the following example describes a minimal graph in natural language form:

```
{
  "nodes": [
    {"node_id": "n1", "task_type": "answer", "content": "<query>"},
    {"node_id": "n2", "task_type": "solve", "content": "<subtask>"},
    {"node_id": "n3", "task_type": "search", "content": "<subtask>"},
  ],
  "edges": [
    {"from": "n2", "to": "n1", "relationship": "solve subtask"},
    {"from": "n3", "to": "n1", "relationship": "provide information"},
  ]
}
```

Here, n1 represents an answering task defining the final research objectives, while n2 and n3 correspond to problem-solving and retrieval subtasks, respectively. The edges indicate dependencies: n2 supports the final answer, and n3's retrieval is guided by the objective of n1.

By formalizing the research process in this manner, we enable the agent not only to generate execution plans but also to reason over the structural dependencies among subtasks, ensuring coherence and systematicity throughout the deep research workflow.

## 2.2 KNOWLEDGE FLOW PLANNER

A high-quality Knowledge Flow is essential for the effective execution of complex research tasks. Rather than constructing the entire structure in a single step, which can lead to instability and reduced control, we employ a Knowledge Flow Planner process that incrementally initializes the flow.

Let $G_t^{init} = \{V_t^{init}, E_t^{init}\}$ be the flow in the $t$-th initialization iteration. Specifically, $G_0^{init} = \{\{v_{query}\}, \emptyset\}$ only contains the query node at the beginning. At each iteration $t$, an LLM planner examines the nodes in the current flow $G_t^{init}$ to identify those requiring further decomposition or additional context. For each node that requires decomposition, the planner generates a set of successor nodes representing sub-questions, intermediate reasoning steps, or supporting evidence for

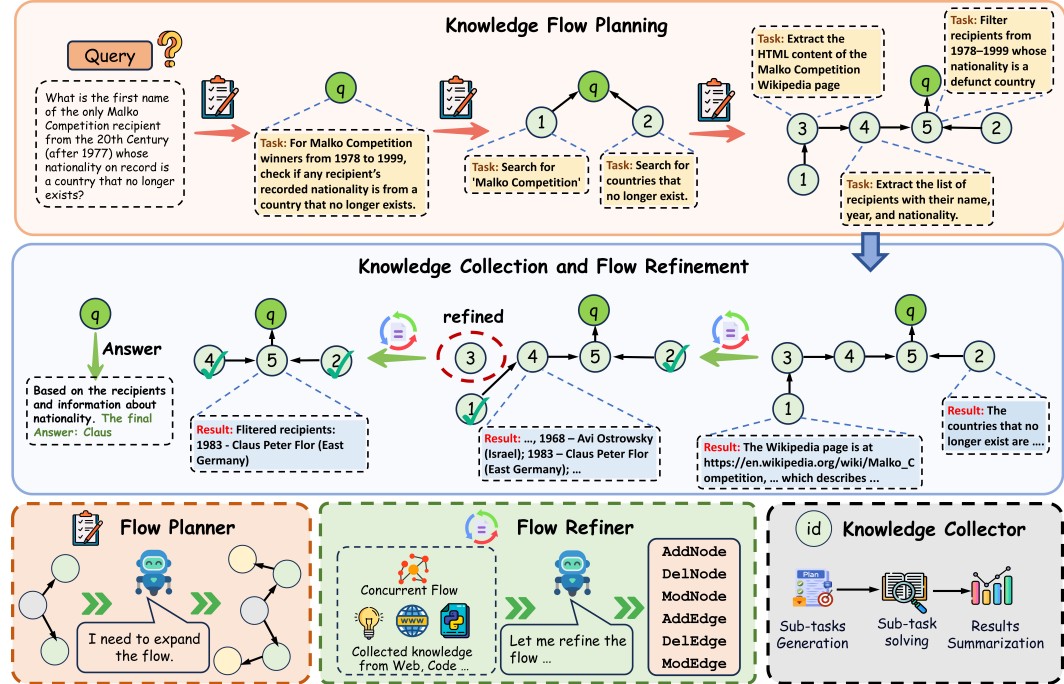

Figure 2: Overview of InternAgent-DR. **Top part** illustrates the Knowledge Flow Planning process, where the Knowledge Flow Planner incrementally expands the structured knowledge flow. **Middle part** depicts the iterative process of Knowledge Collection and Flow Refinement, where nodes are executed by the Knowledge Collector and the flow is dynamically adjusted by the Knowledge Flow Refiner based on newly acquired knowledge. **Lower part** highlights the three key components of InternAgent-DR—Flow Planner (left), Flow Refiner (center), and Knowledge Collector (right)—and their collaborative role in enabling systematic, adaptive, and efficient deep research.

it. The corresponding dependency edges are added to the flow to maintain structural coherence and preserve logical consistency, which can be formulated as follows:

$$G_{t+1}^{init} = \{V_{t+1}^{init}, E_{t+1}^{init}\} = f_\theta^{expand}(G_t^{init}), \tag{1}$$

where $f_\theta^{expand}(\cdot)$ is the trainable LLM planner, $V_{t+1}^{init} = V_t^{init} \cup V_t^{add}$ contains newly added nodes and $E_{t+1}^{init} = E_t^{init} \cup E_t^{add}$ contains newly introduced edges (dependencies) connecting nodes in $V_t^{add}$. This iterative expansion progressively extends the boundaries of the research and deepens the level of exploration within the knowledge flow. The process continues until $f_\theta^{expand}(\cdot)$ yields no additional nodes. Upon completion of the expansion phase, an initial flow $G_0 = G_T^{init}$ is instantiated to support subsequent knowledge collector and flow refiner, where $T$ is the iteration steps in the flow expansion stage.

We curated a dataset of 10k examples to fine-tune a large model for the planner, which we term InternPlanner. Each data point is formatted as a dialogue: the input is a textual description of a flow, and the output is either (i) an updated flow obtained by expanding the current flow by one step, or (ii) the unchanged input flow, indicating the termination of the expansion. Further details about the dataset can be found in Appendix D.

After the initial planning of the knowledge flow, InternAgent-DR then enters another iterative loop of Knowledge Collector and Flow Refiner. This iteration continues until the original user query is successfully resolved. Knowledge Collector and Flow Refiner are described as follows.

## 2.3 KNOWLEDGE COLLECTOR

The Knowledge Collector aims at identifying the outermost executable nodes in the flow—those whose dependencies have all been resolved—and assigns each to an executor agent for processing. These agents, implemented as large language models equipped with tools, decompose the subtask

into a sequential execution trajectory, iteratively reasoning and retrieving information to resolve the node. Available tools include web browsing, file downloading, and visual question answering, etc. A complete list of supported tools is provided in the Appendix B.

After the execution of node $v_i$, its execution state $s_i$ (either success or failure) is updated. If the execution succeeds, the resulting knowledge—either retrieved or derived through reasoning—is distilled into a summarized knowledge context $c_i$, which serves as input for the subsequent execution of the nodes that depend on it. Formally, given $G_t = (V_t, E_t)$ in the $t$-th Knowledge Collector and Flow Refiner iteration, the execution of node $v_i$ can be described as:

$$s_i, c_i = f^{exec}(t_i, d_i | \{c_j | (v_j \to v_i) \in E_t\}), \tag{2}$$

where $t_i$ and $d_i$ are the task type and task description of node $v_i$, $f^{exec}(\cdot)$ is the LLM executor with tools depending on the context knowledge $\{c_j | (v_j \to v_i) \in E_t\}$. After the parallel execution of all the outermost executable nodes, a flow refinement will be conducted based on the newly obtained knowledge, which will be detailed as follows.

## 2.4 KNOWLEDGE FLOW REFINER

After completing the execution of nodes and updating the corresponding knowledge in the iteration, InternAgent-DR activates the Knowledge Flow Refiner to improve the structure of the flow. Leveraging the newly acquired knowledge, Knowledge Flow Refiner analyzes the current flow and identify potential structural adjustments, including the addition, removal, or modification of tasks and dependencies. The goal of Knowledge Flow Refiner is to advance the research task in a reflective way and enhance execution efficiency.

The Knowledge Flow Refiner (achieved by an LLM) is prompted to utilize a set of predefined graph transformation operations to modify nodes and edges in the flow based on the knowledge context and execution states of the existing nodes. These operations include:

- **Add Node (`AddNode`)**: Introduce new nodes to capture missing sub-questions, intermediate reasoning steps, or evidence that were not anticipated in the initial flow.
- **Delete Node (`DelNode`)**: Remove nodes that are redundant, irrelevant, or no longer necessary given the updated knowledge.
- **Modify Node (`ModNode`)**: Modify the attributes of current nodes, especially the content of the sub-task.
- **Add Edge (`AddEdge`)**: Create new dependency edges to reflect newly discovered relationships between nodes.
- **Delete Edge (`DelEdge`)**: Remove edges that represent incorrect, obsolete, or redundant dependencies, ensuring a more reasonable graph structure.
- **Modify Edge (`ModEdge`)**: Modify existing edges to correct dependency directions or improve the structure for more efficient execution.

Formally, $G_{t+1} = f^{refine}(\{V_t, E_t\})$, where $f^{refine}$ is an LLM that generates a sequence of graph transformation operations $\mathcal{O} = \{o_1, o_2, \ldots, o_m\}$ and applies them on $G_t = \{V_t, E_t\}$ to obtain the updated flow $G_{t+1}$. Through ongoing adjustments, InternAgent-DR achieves coherent and goal-directed reasoning.

## 2.5 CONCLUSION GENERATION

At the conclusion of all iterations of knowledge collector and flow refiner, only the initial query node remains unexecuted. If the query is a scientific question that can be answered directly and simply, the query node will be executed using the knowledge from the connected nodes. If the query is to create a detailed scientific report, the final query node will gather knowledge from all nodes in the flow, complete the reasoning process, and provide a full report.

# 3 EXPERIMENTS

To comprehensively assess the capabilities of InternAgent-DR, we conduct experiments on a diverse set of challenging benchmarks, ranging from general question answering to scientific deep research.

## 3.1 EXPERIMENTS SETUP

**Evaluation Benchmarks.** We conduct extensive experiments on four challenging benchmarks, including:

- **GAIA** Mialon et al. (2023): a benchmark of real-world questions that require a set of fundamental abilities such as reasoning, multi-modality handling, web browsing, and generally tool-use proficiency. Our results are based on its 165-question validation set.
- **GPQA** (Rein et al., 2024): a benchmark of 448 multiple-choice questions across biology, chemistry, and physics, authored by domain experts to ensure depth and rigor, thereby providing a stringent evaluation of advanced reasoning and scientific knowledge. We use its 198-question GPQA-diamond subset for evaluation.
- **HLE** (Phan et al., 2025): Humanity's Last Exam is a multimodal benchmark consisting of 2,500 questions across mathematics, humanities, and natural sciences. Developed by subject experts, it provides a frontier-level test of academic competence where current LLMs still perform far below human experts.
- **TRQA** (Zhang et al., 2025b): a domain-specific benchmark for therapeutic target discovery. It covers fundamental biology, disease biology, pharmacology, and clinical medicine, providing a systematic evaluation framework for biomedical research agents. We use its 172-question TRQA-lit subset for evaluation.

**Methods of Comparison.** To validate the effectiveness of InternAgent-DR, we compare InternAgent-DR on GAIA, GPQA, HLE and TRQA against both cutting-edge large language models including Qwen3 series model (Yang et al., 2025), Intern-S1 (Bai et al., 2025), Deepseek-R1 (Guo et al., 2025), GPT-o4-mini and GPT-5 (OpenAI, 2025a), and some state-of-the-art deep research agent, including proprietary approaches OpenAI-DR (Deep Research) (OpenAI, 2025b), Gemini-DR (Google, 2024) and Manus (man, 2025), react agentic model WebDancer (Wu et al., 2025), DeepResearcher (Zheng et al., 2025), WebShaper (Tao et al., 2025), and and open-source frameworks MiroFlow (Team, 2025), OWL (Hu et al., 2025) X-Masters (Chai et al., 2025), JoyAgent (Liu et al., 2025), AWorld (Yu et al., 2025), OAgent (Zhu et al., 2025), Skywork (Zhang et al., 2025a), and Origene (Zhang et al., 2025b). In the experiments, we prompt GPT-o4-mini to serve as the Knowledge Flow Planner, Knowledge Collector and Knowledge Flow Refiner in our workflow.

## 3.2 EXPERIMENT RESULTS

Table 1 and Figure 3 presents the performances of InternAgent-DR and its counterparts on GAIA, GPQA, HLE and TRQA. InternAgent-DR consistently achieves state-of-the-art results across all benchmarks, validating the effectiveness of the systematic design of InternAgent-DR.

### 3.2.1 GENERAL QUESTION ANSWERING

**InternAgent-DR achieves state-of-the-art performance among agentic systems.** On GAIA (Table 1), InternAgent-DR (o4-mini) outperforms both closed-source Manus (73.30%) and open-source OWL (69.70%), and shows strong robustness on Level 3 questions (50.00%). These results indicate that its iterative workflow combining knowledge planning, collection, and refinement is particularly effective for multi-hop and compositional reasoning. Its clear advantage over systems like OpenAI DR and MiroFlow further underscores the impact of structured and dynamic workflow design.

**Agentic systems consistently outperform pure LLMs on complex reasoning tasks.** Larger base models like the Qwen series benefit from greater internal knowledge, but remain limited without structured reasoning. Even domain-finetuned models such as Intern-S1 lag behind agentic approaches. For instance, InternAgent-DR (o4-mini) achieves a GAIA score of 76.96%, far surpassing the same model without agency (16.97%), highlighting that structured task decomposition and flow-based execution are more critical than model size alone.

### 3.2.2 MULTI-DISCIPLINARY RESEARCH AND QUESTION ANSWERING

**InternAgent-DR effectively acquires domain-specific knowledge through Knowledge Flow.** On the GPQA-diamond benchmark, InternAgent-DR (o4-mini) achieves 87.37% average accuracy in

Table 1: Performance comparison on GAIA, GPQA-diamond and HLE benchmarks. The best results are **bolded** and the second best results are underlined. Results not reported in the original papers are denoted as " - ".

| Method | Base Model | GAIA val | | | | GPQA-diamond | | | | HLE | |
|---|---|---|---|---|---|---|---|---|---|---|---|
| | | Level 1 | Level 2 | Level 3 | Avg. | Bio | Chem | Phys | Avg. | text only | All |
| *No Agency* | | | | | | | | | | | |
| Qwen-3-8B | - | 11.32 | 2.32 | 0.00 | 4.85 | - | - | - | 44.44 | - | - |
| Qwen3-32B | - | 13.21 | 3.49 | 3.84 | 6.67 | - | - | - | 49.49 | - | - |
| Qwen3-235B | - | 15.09 | 3.49 | 3.84 | 7.27 | - | - | - | 47.47 | 9.18 | 8.60 |
| Intern-S1 | - | 28.30 | 9.30 | 7.69 | 15.15 | **89.47** | 59.49 | 93.02 | 78.26 | 8.90 | 8.30 |
| Deepseek-R1 | - | 33.96 | 13.95 | 3.84 | 18.78 | 63.16 | 76.34 | 91.86 | 82.32 | 8.60 | - |
| o4-mini | - | 28.30 | 12.79 | 7.69 | 16.97 | 78.95 | 63.44 | 94.19 | 78.28 | 14.50 | 14.28 |
| GPT-5 | - | - | - | - | - | 84.21 | 76.34 | 95.35 | 85.35 | 25.85 | 24.76 |
| *Close-sourced Agentic Framework* | | | | | | | | | | | |
| OpenAI DR | - | 74.29 | 69.06 | 47.60 | 67.36 | - | - | - | - | - | 26.60 |
| Manus | - | 86.50 | 70.10 | **57.70** | 73.30 | - | - | - | - | - | - |
| Gemini DR | - | - | - | - | - | - | - | - | - | - | 26.90 |
| *React Agentic Model* | | | | | | | | | | | |
| WebDancer | QwQ-32B | 61.5 | 50.0 | 25.0 | 51.5 | - | - | - | - | - | - |
| DeepResearcher | Qwen2.5-7B | 24.53 | 18.60 | 3.84 | 18.18 | - | - | - | - | - | - |
| WebShaper | Qwen2.5-72B | 69.2 | 63.4 | 16.6 | 60.1 | 47.37 | 52.69 | 81.40 | 64.65 | - | - |
| *Open-sourced Agentic Framework* | | | | | | | | | | | |
| MiroFlow | Claude 3.7 | - | - | - | 74.50 | - | - | - | - | 29.50 | 27.20 |
| OWL | Gemini 2.5 Pro | 84.90 | 68.60 | 42.30 | 69.70 | 57.89 | 61.29 | 86.05 | 71.72 | - | - |
| X-Masters | Deepseek-R1 | - | - | - | - | - | - | - | - | **32.10** | 27.72 |
| JoyAgent | Claude 4 | 86.8 | 77.9 | 42.3 | 75.2 | - | - | - | - | - | - |
| AWorld | Gemini 2.5 Pro | - | - | - | 67.89 | - | - | - | - | - | - |
| OAgent | Claude 3.7 | 77.36 | 66.28 | 46.15 | 66.67 | - | - | - | - | - | - |
| Skywork | o4-mini | 81.13 | 72.1 | 30.77 | 68.48 | 63.16 | 60.22 | 87.21 | 72.22 | - | - |
| *InternAgent-DR* | | | | | | | | | | | |
| InternAgent-DR | Qwen3-235B | 69.81 | 60.47 | 30.77 | 58.79 | 63.16 | 58.06 | 75.58 | 66.16 | 15.04 | 14.84 |
| InternAgent-DR | o4-mini | **90.56** | **76.74** | 50.00 | **76.96** | 84.21 | **79.57** | **96.51** | **87.37** | 31.60 | **30.80** |

Table 2: Ablation study on the impact of structured planning and refinement. We compare the workflow with conventional sequential planner, the flow planner, and the flow refiner. A checkmark (✓) indicates the component is used. Results are reported on GAIA and GPQA.

| Sequential Planner | Flow Planner | Refiner | GAIA | | | | GPQA | | | |
|---|---|---|---|---|---|---|---|---|---|---|
| | | | Level 1 | Level 2 | Level 3 | Avg | Bio | Chem | Phys | Avg |
| ✓ | – | – | 67.92 | 55.81 | 23.07 | 55.76 | 57.89 | 54.84 | 88.37 | 71.21 |
| – | ✓ | – | 73.58 | 63.95 | 30.77 | 61.82 | 57.89 | 59.14 | 89.53 | 73.74 |
| – | ✓ | ✓ | **90.56** | **76.74** | **50.00** | **76.96** | **84.21** | **79.57** | **96.51** | **87.37** |

Biology, Chemistry, and Physics—outperforming GPT-5 and Intern-S1. This underscores the advantage of dynamic retrieval in accessing context-relevant knowledge, which enables more accurate and flexible scientific reasoning than relying solely on static information.

**General-purpose tools guided by Knowledge Flow can outperform specialized systems.** On the HLE benchmark, InternAgent-DR (o4-mini) achieves the highest accuracy (30.80%), surpassing closed-source systems like OpenAI DR (26.60%) and Gemini Deep Research (26.90%). On TRQA, it reaches 77.9%, outperforming domain-specific agent Origene (60.1%) and scientific model Intern-S1 (49.4%). These results show that a well-structured general-purpose agent can effectively tackle complex scientific and cross-domain tasks.

## 3.3 ABLATION STUDIES

**Ablation on key components.** We conduct ablation studies on two critical components of InternAgent-DR: the Knowledge Flow Planner and the Knowledge Flow Refiner. As shown in Table 2, replacing conventional sequential planner reasoning with our structured Knowledge Flow

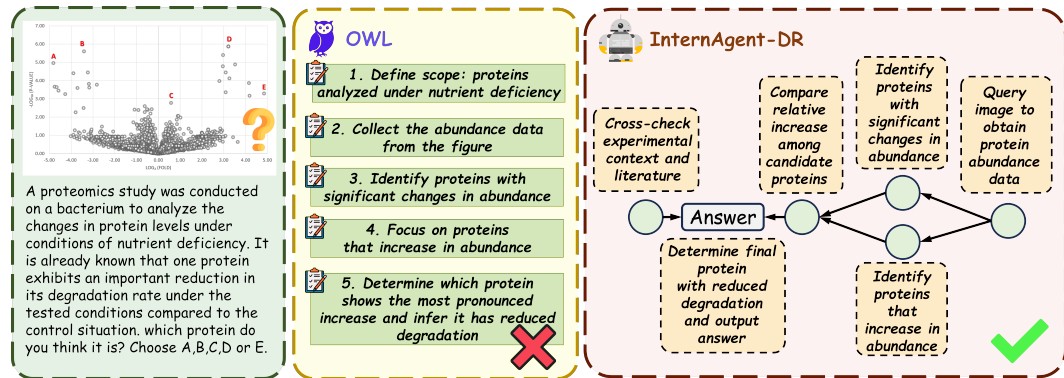

Figure 4: Case study comparing the conventional deep research framework OWL with our InternAgent-DR on a scientific question.

leads to substantial performance improvements, with gains of 6.06% on GAIA and 2.53% on GPQA. This highlights its effectiveness in capturing complex task dependencies and enhancing problem-solving capabilities. Moreover, incorporating the Flow Refiner yields further notable improvements, indicating that dynamic flow refinement enables more flexible task adaptation and strengthens the agent's overall research competence.

**Ablation on trained flow planner.** We conduct experiments with different Flow planners to assess their impact on InternAgent-DR's performance. As shown in Table 3, comparing Qwen3-8B and Qwen3-32B reveals a clear trend: stronger base models produce higher-quality Knowledge Flows, which in turn lead to better overall performance. Moreover, our InternPlanner, finetuned from Qwen3-8B and Qwen3-32B, consistently outperforms their original counterparts, demonstrating both the critical role of the planner and the effectiveness of our training strategy.

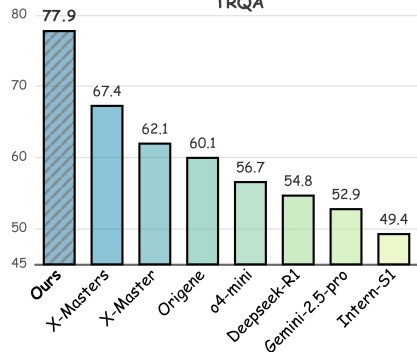

Figure 3: Performance on the TRQA benchmark. InternAgent-DR (Ours) significantly outperforms previous works.

### 3.4 CASE STUDY AND VISUALIZATION

Figure 4 illustrates the contrast between our knowledge-flow-based InternAgent-DR and the conventional sequential planning paradigm, represented by OWL (Hu et al., 2025), in addressing a scientific question. As shown in the figure, OWL decomposes the query into a linear sequence of subtasks—such as understanding, information collection, and identification—that are executed in order. While this pipeline is straightforward, it lacks mechanisms to preserve and integrate intermediate insights, which leads to the dilution of valuable evidence as the chain grows longer.

In comparison, InternAgent-DR constructs a structured knowledge flow directly from the user query, explicitly modeling dependencies between subtasks—for instance, asking a question about an image only after extracting information from it. Each node both executes its designated operation and summarizes its outcome, passing structured intermediate results to subsequent steps along the flow. This design enables selective reuse of prior knowledge, limits the propagation of irrelevant information, and preserves critical evidence throughout the reasoning process.

## 4 RELATED WORK

### 4.1 AGENTIC SYSTEMS

Agentic systems with LLM have evolved from static prompting to perception–action loops, enabling systems to plan (Wang et al., 2023b), act (Yao et al., 2023b), and learn using external tools. Foundational approaches, such as interleaved reasoning–acting frameworks (Yao et al., 2023b) and tree

Table 3: Ablation study on the planner model. We compare various flow planners, including the Qwen3 series and our finetuned InternPlanner. Results are reported on the GAIA benchmark.

| Planner | GAIA | | | |
| --- | --- | --- | --- | --- |
| | Level 1 | Level 2 | Level 3 | Avg |
| Qwen-3-8B | 58.49 | 46.51 | 11.54 | 44.85 |
| InternPlanner-8B (ours) | 70.25 | 67.44 | 34.61 | 66.06 |
| Qwen-3-32B | 77.36 | 67.44 | 30.77 | 64.81 |
| InternPlanner-32B (ours) | **84.91** | **70.93** | **42.31** | **70.91** |

search planning (Yao et al., 2023a), improve reliability in multi-step tasks, while reflective self-revision mechanisms (Shinn et al., 2023) and external memory (Wang et al., 2023a) enhance long-horizon consistency. Recent efforts like OpenHands (Wang et al., 2025) and OpenDevin (Wang et al., 2025) further expand agents' action spaces and mitigate hallucinations through grounded APIs, tool calling, and software-execution platforms, and evaluate them on more realistic interactive benchmarks including AgentBoard (Ma et al., 2024), StuLife (Cai et al., 2025), and SWE-bench Verified (bench Team, 2024). Multi-agent orchestration involves role-specialized collaboration and negotiation (Zhuge et al., 2024; Qiu et al., 2025), replacing single-agent end-to-end optimization with a modular and scalable approach.

Despite these advances, most general-purpose agents target short to medium horizon tasks and inter-active environments. Scientific research (OpenAI, 2025b), however, requires handling long-horizon workflows, integrating diverse knowledge, and adapting strategies dynamically. This motivates the development of research-oriented agents, which focus on structured, adaptive, and knowledge-driven scientific inquiry.

### 4.2 DEEP RESEARCH AGENTS

Recent advances in Deep Research (DR) agents extend LLMs from retrieval-augmented generation to dynamic, tool-driven research workflows. Early systems such as WebGPT (Nakano et al., 2021) and Toolformer (Schick et al., 2023) explored web and API integration, while recent industrial solutions (e.g., OpenAI DR (OpenAI, 2025b), Gemini DR (Google, 2024), Grok DR (xAI, 2025), Perplexity DR (Perplexity, 2025)) incorporate adaptive planning, iterative retrieval, and multimodal reasoning. A key distinction is between static pipelines (e.g., AI Scientist (Lu et al., 2024), Agent Laboratory (Schmidgall et al., 2025), and InternAgent (Team et al., 2025)) and dynamic workflows, where plans evolve during execution. Dynamic workflows further divide into single-agent systems (Search-o1 (Li et al., 2025a), WebDancer (Wu et al., 2025), Qwen DeepResearcher (Qiao et al., 2025)) that unify reasoning and tool use, and multi-agent systems (OpenManus (Project, 2025), OWL (Hu et al., 2025), AWorld (Yu et al., 2025)) that distribute subtasks for parallel and specialised execution. While single-agent designs enable end-to-end optimization, multi-agent architectures offer modularity and scalability—crucial for complex research.

Recent studies also highlight graph-structured retrieval and adaptive workflows (e.g., GeAR (Shen et al., 2024), PANGU DeepDiver (Shi et al., 2025), Alita (Qiu et al., 2025)), showing the benefit of explicit structures and self-evolving mechanisms for multi-hop reasoning. However, the existing DR agents still suffer from sequential bottlenecks and limited hierarchical decomposition, moti-vating frameworks like our InternAgent-DR that integrate multi-agent coordination with dynamic structured knowledge flow.

## 5 CONCLUSION

In this work, we introduce **InternAgent-DR**, a multi-agent deep research system built on a dynamic structured knowledge flow. By explicitly modeling dependencies among subproblems and key con-cepts, the system enables both deep reasoning within local regions of the knowledge flow and coher-ent knowledge propagation at a global level. The dynamic flow framework allows InternAgent-DR to iteratively plan, expand, and refine research workflows, supporting hierarchical task decomposi-tion, parallel exploration, and adaptive strategy adjustment based on intermediate findings. These re-

sults highlight the effectiveness of combining structured knowledge flow planning with multi-agent orchestration, suggesting that such frameworks offer a promising direction for building autonomous, reflective, and scalable systems capable of tackling complex scientific research tasks.

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

## A    USE OF LLMS

Our use of large language models (LLMs) is limited to assisting with manuscript drafting and refinement to ensure clarity and coherence.

## B    TOOLS OF KNOWLEDGE COLLECTOR

We provide a set of tool wrappers used by the Knowledge Collector. Each tool is designed with concurrent safety, creating independent toolkit instances to avoid state conflicts. Table 4 summarizes the available tools.

Table 4: The tools in Knowledge Collector

| Tool | Purpose |
| --- | --- |
| search_google | Use Google search engine to search information for the given query |
| search_wiki | Search the entity in WikiPedia and return the summary of the required page, containing factual information about the given entity |
| search_wiki_revision | Search Wikipedia to get the latest Wikipedia revision *at or before* the end of the given (year, month) |
| search_archived_webpage | Given a url, search the wayback machine and returns the archived version of the url for a given date |
| extract_document_content | Extract the content of a given local document and return the processed text. It can process various types of documents, including text, image, table, audio, video, zip, json, xml, pdf, py etc |
| extract_url_content | Extract the html content of a given url and return the processed text |
| ask_question_about_image | Answer image questions with optional custom instructions |
| ask_question_about_audio | Ask a question about the audio and get the answer using multimodal model |
| ask_question_about_video | Ask a question about the video using Gemini multimodal capabilities |
| download_media_from_url | Download any given URL (image, video, audio, document, or webpage) |
| execute_code | Execute a given code snippet |
| browse_url | A powerful toolkit which can simulate the browser interaction to solve the task which needs multi-step actions |
| ocr2text | OCR the image and return the text |

## C    SUMMARIZER

A summarizer has been developed to generate conclusions for the answer node, featuring two modes of operation.

**Question-answering tasks**: When the objective is to answer a specific question, the task usually requires a strict logical progression of reasoning. In such cases, the later nodes in the execution graph—particularly solve nodes—tend to encapsulate the reasoning steps that are most directly related to the final answer. By contrast, earlier nodes such as search nodes often contain intermediate evidence or auxiliary information that, while necessary for the reasoning process, does not itself contribute to the correctness or clarity of the final response. To enhance efficiency and reduce noise, the summarizer in this mode selectively incorporates only the dependent predecessor nodes of the final answer. This targeted approach ensures that the summary remains focused, concise, and aligned with the logical chain that directly supports the solution. The main benefit is an improvement in answer precision and interpretability, as irrelevant or redundant details are filtered out.

**Report-generation tasks**: By contrast, when the task involves producing a comprehensive report, the goal is not merely accuracy but also coverage and richness. In this context, limiting the summarizer to dependent nodes would risk omitting potentially valuable background, context, or supporting evidence. Therefore, for report generation, the entire execution graph produced by InternAgent-DR is passed to the summarizer. This design allows the system to synthesize information from all nodes—including search, solve, and answer stages—so that the final report captures not only the

core reasoning steps but also the broader landscape of evidence. The benefit of this approach is that the generated report provides a more holistic view of the research process, offering readers both the conclusions and the supporting context, which increases transparency, interpretability, and informational richness.

**Advantages of the dual-mode design**: This bifurcated summarization strategy balances efficiency with completeness. For question answering, it minimizes cognitive load and reduces error propagation by concentrating only on essential reasoning chains. For report writing, it maximizes informativeness and ensures that potentially useful evidence is not prematurely discarded. Together, these modes enable InternAgent-DR to flexibly support both precise problem-solving and broad knowledge synthesis, depending on the user's research goals.

## D   DATASET AND TRAINING

We employ knowledge distillation from GPT-O4-mini to train our InternPlanner. Specifically, for the training data, we collect a set of Wikipedia entries that inherently exhibit entity dependencies. These dependencies are extracted and organized into structured entity graphs. The entity graphs are then fed into O4-mini, which generates questions for each node based on its corresponding dependencies and subsequently integrates these questions into a single summary question, serving as the user query in our dataset. For the graph generation process, the obtained entity graphs are directly converted into a natural language representation according to our predefined format, which is included as the assistant output in the labeled dataset. A detailed description of the data format is provided in Box D.

During training, the labeled data is decomposed into multiple single-turn question-answer pairs, which are then used to fine-tune Qwen3-series models via supervised fine-tuning (SFT).

### D. Data Format

```
{
  "messages": [
    {
      "role": "user",
      "content": "You are a graph planner agent. Your task is to
          decompose any user question into a logical graph of tasks,
          and iteratively refine the graph when node knowledge
          becomes available.

        ### Example Input Graph
        {
          "nodes": [
            {"node_id": "n1", "type": "answer", "task": "Explain why
                sugar-free drinks can still contain carbohydrates"}
          ],
          "edges": []
        }

        Input graph:
        <generated_input_graph>"
    },
    {
      "role": "assistant",
      "content": "<generated_output_graph>"
    }
  ]
}
```

# E  ADDITIONAL CASE STUDIES

## E.1  QUESTION ANSWERING

Below we choose one question from GAIA level 2 to show our execution logic. The content of the question is **"What is the latest chronological year date in the image from the first citation of Carl Nebel's Wikipedia page (Aug 2023)?"**. This question is inherently *logic-intensive*: the answer is not present on the query page but must be derived through a chained, evidence-preserving procedure.

Our InternAgent-DR solves this question step by step, in a very logical order. Starting from the revision-resolved entry point, the planner instantiated a dependency-aware, tool-grounded pipeline: `n1` resolves the August 2023 Wikipedia revision; `n2` extracts the first citation URL; `n3` fetches the citation page HTML; `n8` identifies and downloads the first in-article image; `n6` performs OCR over the downloaded image; and `n7` parses four-digit years to determine the latest date. This design operationalizes a search–extract–process–analyze flow where each step produces verifiable intermediate artifacts (URLs, HTML snapshots, files, OCR text) that can be audited, cached, and reused. By encoding dependencies in a graph, InternAgent-DR enables deterministic, provenance-preserving re-execution, isolates errors to specific nodes, and supports targeted recovery without rerunning the entire pipeline—yielding stronger reproducibility, interpretability, and multi-tool coordination than monolithic, end-to-end prompting. Table 5 shows every node in our InternAgent-DR graph system, including task content, type, status and its output(part).

Since this question requires each step to rely strictly on the results of the previous one, the overall process forms a linear execution flow. The execution followed a strict topological ordering:

$$n1 \rightarrow n2 \rightarrow n3 \rightarrow n8 \rightarrow n6 \rightarrow n7 \rightarrow task$$

## E.2  REPORT GENERATION

The following report was generated by our InternAgent-DR system to answer the query **"Help me research the latest progress in multi-agent AI scientists in 2025"**. The planner decomposed the original research query into a set of interconnected subtasks, enabling systematic information gathering, reasoning, and synthesis. The resulting execution graph consisted of 7 nodes spanning three major categories—search, solve, and answer—which together captured the full problem-solving workflow. Table 6 provides a structured overview of all nodes and their corresponding roles within the pipeline. Importantly, in the case of report generation, the graph structure highlights its advantages even more clearly, since the dependencies across different report sections are relatively weak, allowing parallel execution to be fully leveraged. Therefore, we can generate the report below in 10 minutes. An example output report is shown below.

The execution followed a topological ordering, where nodes grouped together indicate parallel execution:

$$n3, n4s, n7 \rightarrow n2, n4, n6 \rightarrow task$$

Table 5: Execution Trace of InternAgent-DR(Question Answering Case Study)

| Node | Task | Type | Status | Output(Part) |
|------|------|------|--------|--------------|
| n1 | Use `search_wiki_revision` to get Carl Nebel Wikipedia revision (Aug 2023) | search | Success | The Carl Nebel Wikipedia page revision dated 2023-08-05T13:53:28Z carries oldid 1168855983 and is accessible at `https://en.wikipedia.org/w/index.php?title=Carl_Nebel&oldid=1168855983` |
| n2 | Open Carl Nebel revision and extract first citation URL from References | solve | Success | The first citation in the References section is 'Thieme–Becker', entry "Nebel, Carl," with URL `https://de.wikipedia.org/wiki/Thieme-Becker` |
| n3 | Use `extract_url_content` to fetch citation page HTML | search | Success | Extracted the HTML content of the German Wikipedia page at `https://de.wikipedia.org/wiki/Thieme-Becker`, which presents Thieme-Becker as a German biographical dictionary of artists... |
| n8 | Extract first image from citation page and download locally | search | Success | The first image in the 'Thieme-Becker' article body has source URL `https://upload.wikimedia.org/wikipedia/commons/thumb/c/c5/Perwanger%2C_Chr...` |
| n6 | Apply `ocr2text` on downloaded image | solve | Success | The OCR2Text tool returned success and extracted the following lines of text: Pervinquiere-Pescatori; Pervinquiere, Henri Baron, Tiermaler,; Perz, Michael, Stukkator, tatig 1701 im; 1883 Poitiers; Bild im Mus. ebda.; Rathaus zu Landsberg a. Lech... |
| n7 | Parse OCR text to extract all 4-digit years and find latest | solve | Success | From the OCR-extracted text, the unique four-digit years identified are [1558, 1577, 1610, 1645, ... 1913, 1915, 1924, 1925, 1927], and the latest chronological year among these is 1927. |
| task | What is the latest chronological year date in the image from the first citation of Carl Nebel's Wikipedia page (Aug 2023)? | answer | Success | 1927 |

Table 6: Execution Trace of InternAgent-DR(Report Generation Case Study)

| Node ID | Type | Task Description |
|---------|------|-----------------|
| task | answer | Help me research the latest progress in multi-agent AI scientists in 2025. |
| n2 | solve | [Background & Methods] Synthesize definitions, scope, historical context, and classify core methods |
| n3 | search | [Background & Methods] Collect definitions, seminal works, and representative methods |
| n4 | solve | [Datasets/Applications] Summarize datasets, benchmarks, evaluation metrics, and applications |
| n4s | search | [Datasets/Applications] Collect datasets, benchmarks, evaluation results, and application examples |
| n6 | solve | [Challenges & Future Work] Analyze challenges, limitations, and outline future directions |
| n7 | search | [Challenges & Future Work] Collect discussions of current challenges, limitations, and future outlook |

# F  PROMPTS

In this section, we show the prompts of each agents, to clarify the workflow.

**Flow Planner Prompt**

You are a graph planner agent.
Decompose any user question into a logical graph of tasks, refining iteratively when node knowledge becomes available.

Output strictly JSON with "nodes" and "edges".

Node rules:
- nodes:
  - "node_id": unique id (e.g., "n1")
  - "type": ["search", "solve", "answer"]
  - "task": short natural language description
- edges:
  - "from", "to", "relationship"

Node type:
- search: collect raw info
- solve: reason, compute, integrate, or handle complex tasks
- answer: final solution (only one)

Refinement:
- Break nodes into concrete child tasks and connect edges
- Add edges if a node depends on another's knowledge
- Modify incorrect/unreasonable tasks
- Expand only one layer per iteration
- Stop and output "Perfect!" if all nodes are concrete and complete, and no further decomposition is possible
- For survey/review questions, ensure major aspects/perspectives are covered

Input format:
- JSON graph with initial "answer" node
- Do not modify answer node
- Always produce valid JSON
-  If nodes > max nodes, do not add new nodes, clarify existing ones; if clear, output "Perfect!"

Example behavior:
- If all nodes concrete and sufficient → "Perfect!"
- Otherwise, add one layer of concrete child nodes

### Example

**Input graph:**
```json
{{
  "nodes": [
    {{"node_id": "n1", "type": "answer", "task": "Explain why sugar-free drinks can still contain carbohydrates"}}
  ],
  "edges": []
}}
```
Make sure to finish your plan in {max_iter} turns, and this is your {current_iter} turn.

Make sure to not add more than {max_nodes} nodes.

This is the input graph {graph} to answer the question{question}

**Flow Refiner Prompt**

You are a Graph Reasoning Agent managing and updating DAG task graphs for multi-step workflows.

You are given a graph and a query, and need to modify the graph to answer the query.

### Input
1. **graph**: JSON with
   - `nodes`: each with `node_id`, `status` (pending/executed), `task`, `type` (search/solve/answer), `final_response`, `success`, `reasoning`
   - `edges`: each with `from`, `to`, `relationship`
2. **query**: overall problem the graph solves

Only pending nodes can be modified. Do not modify executed nodes or the answer node.

### Allowed Actions
- Node: `add_node`, `remove_node`, `modify_node`
- Edge: `add_edge`, `remove_edge`, `modify_edge`

### Modification Rules
- Refine unclear tasks, remove redundant nodes, add nodes for alternative execution paths.
- Add/remove edges to maintain correct dependencies.
- Keep the graph connected, acyclic, and minimal.
- Only modify nodes/edges needed to fix failures.
- If no changes are needed, output `[]`.
- Do not change the answer node.

### Output Format
JSON array of modifications. Each modification includes:
- `action`: one of allowed actions
- `node_id` / `from_node` / `to_node`
- `attributes`:
  - Node: `{ "task": "...", "type": "search|solve|answer"}`
  - Edge: `{ "relationship": "..." }`
- `reason`: concise explanation

### Example Output
```json
[
  {
    "action": "add_node",
    "node_id": "n6",
    "attributes": {
      "task": "Validate the final answer against multiple sources",
      "type": "solve"
    },
    "reason": "Introduce an explicit validation step to improve reliability"
  },
  {
    "action": "add_edge",
    "from_node": "n3",
    "to_node": "n6",
    "attributes": { "relationship": "produces draft answer" },
    "reason": "The output of n3 should flow into the new validation step"
  },
  {
    "action": "add_edge",
    "from_node": "n6",
    "to_node": "n4",
    "attributes": { "relationship": "validated answer" },
    "reason": "Ensure the validated result is passed downstream to n4"
  }
]
```
This is the input graph:

=====================================
{graph}
=====================================

This is the query that the graph is meant to solve:

=====================================
{query}
=====================================

**Knowledge Collector Prompt**

You are a Task Graph Agent. Your job is to:
1) Decompose the given task into minimal, concrete subtasks.
2) Execute subtasks using available tools or upstream knowledge.

### Inputs
- **task**: current task description
- **query**: overall question
- **Full subtask list**: history subtask list
- ** subtask **: Current subtask to execute
- **tools**: available execution tools

### Subtask Planning Rules
- Split the task into as few subtasks as possible (<{max_subtasks}>), unless it is a survey/review task, then cover major aspects.
-   Subtasks describe **what to do**, not how. No tools, parameters, or implementation details in subtask text.
- Return subtasks as:
<tasks>
<task>Subtask 1</task>
<task>Subtask 2</task>
</tasks>

### Execution Rules
- Decide for each subtask whether to use tools or answer directly.
- Always use appropriate tools for knowledge retrieval, code execution, or document processing.
- Include concise, reproducible outputs; list files produced.
- Reference upstream `knowledge_info` to avoid redundant work.
- If blocked by missing inputs or errors, return a clear blocker description.
- Do NOT reveal chain-of-thought. Keep outputs machine-readable and self-contained.

### Output
- For planning: numbered list of subtasks as above
- For execution: structured JSON with results, artifacts, and any notes on blockers
"""
Overall task:
==============================
{task}
==============================

The task serves as a part to solve this question(- This only tells you what the overall question is.  - You are NOT asked to solve the query directly.):
==============================
{query}
==============================
Full subtask list (from the planner), ordered, some subtasks are already completed:
==============================
{history_subtasks}
==============================

Current subtask to execute (one item from the list above and this is your current subtask):
==============================
{subtask}
==============================
Here are the available tools

==============================
{tools}
==============================

**The following content is an example of a report generated by InternAgent-DR:**

# Multi-Agent AI Scientists in 2025

## INTRODUCTION

With the rapid development of Large Language Models (LLMs) and Multi-Agent System (MAS) technologies, 2025 has seen the emergence of a series of "multi-agent AI scientist" platforms capable of collaboratively carrying out the entire process of scientific research. This report aims to provide a comprehensive reference for researchers and practitioners by synthesizing the latest definitions, historical background, core methodologies, datasets and evaluation benchmarks, real-world application cases, as well as the challenges faced and directions for future development in the field of multi-agent AI scientists in 2025.

## OVERVIEW OF METHODS / WORKFLOWS

This report is based on a comprehensive analysis of multi-node workflows, drawing mainly on three core sections: (i) background and methods review, (ii) datasets and applications summary, and (iii) analysis of challenges and future directions. Each section compiles key insights from academic papers, industry blogs, white papers, technical reports, and authoritative surveys, preserving original references and data rigorously.

## FINDINGS AND SYNTHESIS

### 1. BACKGROUND AND METHODS

#### 1.1 DEFINITIONS

- **Multi-Agent System (MAS)** — a computerized system composed of multiple interacting agents (software, robots, humans, or hybrids). Core features include autonomy, local views (no global state), and decentralization. Typical research themes encompass agent communication and coordination, distributed problem solving, multi-agent learning, DCOPs, BDI architectures, and LLM-based MAS.

- **Multi-Agent "AI Scientist"** — a specialized MAS in which agents collaborate to complete the end-to-end scientific research process: hypothesis generation, experimental design, data interpretation, manuscript writing, and peer review. These systems are built upon advances in LLMs: each agent is usually an LLM or LLM+tools, specialized for a specific subtask of the scientific method.

#### 1.2 HISTORICAL DEVELOPMENT AND BACKGROUND

**(a) Milestones in Laboratory Automation**

- **1875**: First reports of automated scientific equipment, custom-built by scientists to address lab problems.

- **Post–WWII**: Commercial vendors began offering increasingly complex automated lab equipment.

- **1981–1983**: Dr. Masahide Sasaki established the first fully automated laboratory.

- **1993**: Dr. Rod Markin developed the first clinical laboratory management system and led CTASSC (Clinical Testing Automation Standards Steering Committee).

- **2004**: The NIH Roadmap emphasized molecular libraries and imaging technologies, driving large-scale automation in biomedical research.

- **Mid-2010s**: Rise of commercial "on-demand" remote labs (e.g., Emerald Cloud Lab, Strateos); studies showed that over 90% of methods in biomedical papers could be accessed through such services.

- **Low-cost automation era**: Scripting languages (e.g., AutoIt) and open-source modules (LEGO, 3D printers) lowered the barrier for small labs.

**(b) Early "Robot Scientist" Prototypes**

- **Adam (Robot Scientist)** by Ross King et al. — the first system to autonomously discover new knowledge in yeast functional genomics. Capabilities included hypothesis generation, experimental design, robotic execution, results interpretation, and iterative experimentation (domain: *Saccharomyces cerevisiae*).

- **Eve** — a successor system focused on automated drug screening and reproducibility testing in cancer research.

**(c) Conceptual Basis: MAS**   Traditional MAS research emphasizes communication, coordination, distributed problem solving, multi-agent learning, DCOPs, and BDI-style architectures.

**(d) Shift to AI-Driven MAS for Scientific Research**   Advances in LLMs have enabled specialized agents to handle scientific subtasks (literature review, hypothesis generation, coding, data analysis, writing, peer review). Pipelines evolved from single tools to multi-agent workflows orchestrated by a central controller, simulating the iterative cycle of the scientific method. Human intervention is minimized: humans primarily handle wet-lab work and final oversight, while AI agents manage creativity, design, execution, analysis, and writing.

**(e) Representative Prototypes and Frameworks before 2025**

- **"Lab in the Loop" (FutureHouse & Oxford) — System: Robin**. Agents: Crow & Falcon (literature analysis; Crow summarizes topics and proposes experimental designs; Falcon writes extended technical reports), Finch (data analysis of raw data such as RNA-seq and flow cytometry), and a Tournament Judge (hypothesis ranking using Bradley–Terry paired comparisons). Closed-loop iterations proceed as Crow/Falcon → Finch → Judge. In roughly ten weeks, the system completed drug repurposing studies for ten diseases and identified candidate compounds for dry age-related macular degeneration (dry AMD). Humans performed wet-lab experiments and finalized the manuscript.

- **Zochi (Intology)** — a pipeline of retrieval, hypothesis generation, code execution, analysis, writing, and optional automated peer review. Human intervention occurs only ahead of major computation and during final polishing (figures, citations, formatting). Reported outcomes include ACL 2025 acceptance (average reviewer score 4.0), and projects such as CS-ReFT and Tempest/Siege.

- **The AI Scientist (Sakana.ai)** — a four-stage loop: *Idea Generation* (guided by code templates and Semantic Scholar search), *Experimental Iteration* (running and logging code experiments with quantitative/visual outputs), *Paper Write-Up* (LaTeX drafts with auto-citations), and *Automated Peer Review* (LLM-driven near-human feedback). The loop is open-ended (each round is archived and informs the next). Estimated cost is approximately $15 per paper. Known limitations include lack of precise layout/typography control, occasional implementation bugs, and security concerns when running unsandboxed code.

## 1.3 COMMON ARCHITECTURAL THEMES

- **Agent specialization** across subtasks: literature, hypothesis, coding, analysis, writing, and review.

- **Workflow orchestration**: centralized coordination following the scientific method's sequence.

- **Iterative cycles**: hypothesis → experiment → analysis → refinement.

- **Minimal human intervention**: humans focus on wet-lab work and final polishing.

- **Automated quality control**: hypothesis ranking (e.g., Bradley–Terry), LLM-based peer review, and meta-analysis.

## 1.4 EVOLVING STANDARDS AND FRAMEWORKS

- Legacy MAS protocols (e.g., KQML, ACL, FIPA) are increasingly complemented or replaced by LLM-driven frameworks (e.g., CAMEL).

- Safety and ethics considerations include sandboxing, traceability, transparent AI attribution, and biosafety/software-security safeguards.

## 1.5 PIONEERING WORKS

1. **Lu et al., "The AI Scientist: Towards Fully Automated Open-Ended Scientific Discovery"** (arXiv:2408.06292, Sep 2024). First comprehensive LLM-agent framework spanning idea generation, code execution, visualization, manuscript writing, and review loops; estimated cost <$15 per paper; code: `https://github.com/SakanaAI/AI-Scientist`.

2. **Su et al., "Many Heads Are Better Than One: Improved Scientific Idea Generation by an LLM-Based Multi-Agent System"** (arXiv:2410.09403, May 2025; ACL 2025). Introduces Virtual Scientists (VirSci)— a set of specialized LLMs that generate, evaluate, and optimize research ideas—outperforming single-agent baselines.

3. **Ghareeb et al., "Robin: A Multi-Agent System for Automating Scientific Discovery"** (arXiv:2505.13400, May 2025 submission). A closed-loop "lab-in-the-loop" MAS integrating literature review, hypothesis generation, experiment design, data analysis, and iterative refinement.

## 1.6 REPRESENTATIVE METHODS AND MODEL ARCHITECTURES

**(a) System Goals and Benefits**   These systems address dynamic, dependency-rich research tasks; enable parallel exploration; compress and merge context; and, compared with single-agent baselines, can deliver performance gains on the order of $\sim$90.2% at the cost of roughly $15\times$ token usage.

**(b) Orchestrator–Worker Pattern**   A *Lead (Orchestrator) Agent* analyzes user queries, drafts strategies, persists plans, spawns/monitors subagents, synthesizes outputs, and manages loop termination. *Subagents (Workers)* execute specialized tasks in parallel (e.g., web retrieval, data extraction) using interleaved thinking strategies to optimize queries and return structured results. A dedicated *CitationAgent* can post-process documents to produce precise citations.

**(c) Model Backends and Tool Interfaces**   Representative configurations include a powerful model for the lead agent (e.g., Claude Opus 4) and more efficient models for subagents (e.g., Claude Sonnet 4). Tool access is mediated by the Model Context Protocol (MCP) for network search, document retrieval, and workspace APIs. Dynamic multi-step retrieval often replaces static RAG, guided by in-prompt tool-selection heuristics.

**(d) Prompt-Engineering Principles**

1. Simulate agent execution to expose prompt failure modes.
2. Teach the orchestrator to delegate: clarify goals, output schemas, and tool guidance.
3. Scale the number of agents and tool calls with query complexity.
4. Design clean, purpose-built tools.
5. Let agents diagnose and improve other agents' prompts and tool use.
6. Start broad, then progressively focus query strategies.
7. Use extended thinking as an explicit scratchpad.
8. Parallelize tool calls to reduce latency by up to 90%.

**(e) Evaluation Strategies**   Small-sample trials ($\sim$20 queries) assess prompt and tool efficacy; LLM-as-judge scoring covers factuality, citation accuracy, completeness, source quality, and efficiency. Human evaluation supplements these metrics to surface hallucinations and biases. Outcome-centric evaluations focus on persistent state changes.

**(f) Production Engineering and Reliability**   Stateful execution, checkpointing and retries; full tracing of agent decisions and tool calls; progressive ("rainbow") deployment; external storage for plan digests and artifacts; and artifact systems to reduce token costs.

**(g) Architectural Challenges and Future Directions**   Synchronizing subagents remains a bottleneck (driving interest in asynchronous coordination). High operational costs encourage focusing on high-value domains. Coupling across subtasks and global-context management also remain open challenges.

## 2. DATASETS, BENCHMARKS, AND APPLICATION CASES

### 2.1 DATASETS

**(a) AI Idea Bench 2025 (arXiv:2504.14191)**   A large-scale benchmark for idea generation covering recent venues (ICLR 2025, CVPR 2024, ECCV 2024, NeurIPS 2024, ICML 2024, NAACL 2024, EMNLP 2024, ACL 2024). It aggregates 3,495 target AI papers (post–Oct 10, 2023), plus the five most-cited works for each. Tasks include Idea Multiple-Choice Evaluation (IMCQ), Idea-to-Idea (I2I), and Idea-to-Topic (I2T). Representative results are summarized in Table 7.

Table 7: Representative AI Idea Bench 2025 results (illustrative extraction).

| Method | I2T | IMCQ (A/B) | Novelty / Feasibility | Notes |
|---|---|---|---|---|
| AI-Scientist | 5.0/5.0 | 3.591 / 2.734 | $17.003 \times 10^{-3}$ (highest novelty) | — |
| AI-Researcher | 4.994/4.995 | 2.807 / 2.024 | $9.728 \times 10^{-3}$ (overall feasibility) | — |
| VIRSCI | 4.974/4.983 | 2.937 / 2.123 | strongest protocol alignment | — |
| SCIPIP | 4.986/— | 2.437 / — | — | — |

**(b) Open-Source Image Datasets (DatasetAgent; arXiv:2507.08648)**  A multi-agent pipeline where four specialized agents use multimodal LLMs and an image-optimization toolkit to expand or build datasets according to user specifications. Downstream uses include image classification, detection, and segmentation. Specific source datasets (e.g., COCO, ImageNet) are not enumerated here.

## 2.2 BENCHMARKS AND EVALUATION INDICATORS

- **Stanford HAI 2025 AI Index Report**: a general AI benchmark compendium including emerging multi-agent indicators. Highlights include improvements on MMMU (+18.8pp), GPQA (+48.9pp), and SWE-bench (+67.3pp). Responsibility-focused benchmarks covered include HELM Safety, AIR-Bench, and FACTS. The performance gap between open and closed weights narrows from 8% to 1.7%. Multi-agent findings note that language-model agents can surpass humans on time-limited programming tasks.

- **DatasetAgent (arXiv:2507.08648)**: evaluated via downstream CV tasks (classification, detection, segmentation); the abstract reports methodology without detailed aggregate metrics.

## 2.3 REAL-WORLD APPLICATIONS AND CASE STUDIES

**(a) Google AI Co-Scientist**  A Gemini 2.0–based coalition of agents (Generation, Reflection, Ranking, Evolution, Proximity, Meta-review) supervised by a *Supervisor* agent. Capabilities include iterative self-debate and tournament-style ranking (Elo-like), automated literature review, hypothesis generation, proposal writing, experimental design, and recursive self-critique. Demonstrations:

- **AML drug repurposing**: AI proposed KIRA6; validated *in vitro* across multiple AML cell lines, showing significant tumor-suppression at clinically relevant concentrations.

- **Liver fibrosis targets**: identified epigenetic targets with anti-fibrotic activity in human liver organoids (collaboration with Stanford; $p < 0.01$).

- **Antimicrobial resistance mechanisms**: reproduced cf-PICI host-range expansion mechanisms with timelines consistent with unpublished experimental results (Fleming Initiative & Imperial College London).

**(b) Stanford Virtual Lab of AI Scientists**  An AI Principal Investigator agent coordinates subagents spanning immunology, computational biology, and machine learning, along with critic agents. Workflow: a human states a scientific challenge, the AI PI assigns roles, the agents conduct a parallel "lab meeting" lasting seconds to minutes and record meeting minutes, and tools such as AlphaFold assist in experimental design. A case study on SARS-CoV-2 vaccine design selected nanobody candidates that were validated for stability, variant binding, and low off-target effects by Chan Zuckerberg Biohub, demonstrating the acceleration provided by autonomous, high-throughput virtual experimentation.

**(c) PriM: Principle-Inspired Material Discovery (arXiv:2504.08810)**  A language-reasoning multi-agent system combining domain principles to guide exploration of chemical space via hypothesis generation and round-table MAS discussion. Demonstrated higher discovery rates and improved material properties in a nano-helix materials case compared with baselines. Code/data: `https://github.com/amair-lab/PriM`.

**(d) SparksMatter: Autonomous Inorganic Materials Discovery (arXiv:2508.02956)**  A GPT-4–series multi-agent framework (*Scientist*, *Planner*, *Assistant*, *Critic*) following the idea→plan→experiment→report pipeline. Integrated tools include the Materials Project API, MatterGen, MatterSim, and CGCNN. Demonstration tasks:

- **Green thermoelectrics**: proposed Zintl phase $CaMg_2Si_2$ with energy above the convex hull $\leq 0.05\,\mathrm{eV/atom}$; predicted band gap $0.556\,\mathrm{eV}$ and bulk modulus $54.5\,\mathrm{GPa}$, with follow-up validation plans.

- **Soft inorganic semiconductors**: identified $Hg_2MgRb_2$ (bulk modulus $K = 19.94\,\mathrm{GPa}$; band gap $1.52\,\mathrm{eV}$; energy above hull $0.036\,\mathrm{eV/atom}$) including structure and synthesis routes.

- **Lead-free perovskite oxides**: selected $KNaNb_2O_6$ isotypes ( $E_{\mathrm{hull}} < 0.03\,\mathrm{eV/atom}$; band gap $\approx 2.4\,\mathrm{eV}$; bulk modulus $\approx 98\,\mathrm{GPa}$ ) with ferroelectric potential and validation strategies.

Benchmarking indicates superior novelty, depth, and rigor relative to OpenAI o3 / o3-deep-research / o4-mini-deep-research baselines; the authors identify scientific limitations and propose concrete improvements. Code/data: `https://github.com/lamm-mit/SparksMatter`.

## CHALLENGES, LIMITATIONS, AND FUTURE WORK

### 3.1 MAJOR CHALLENGES AND SYSTEMIC RISKS

(IONI.ai Compliance Blog, Feb 15, 2025)

1. **Agent failures**: errors from a single agent can cascade. Mitigation: data-governance policies, pre-deployment testing, and fault isolation.

2. **Coordination complexity**: task allocation and messaging protocols are hard, especially with diverse agent roles. Mitigation: layered, sequential, or bidirectional coordination frameworks.

3. **Unpredictable / emergent behaviors**: decentralized agents may conflict or act unexpectedly. Mitigation: real-time monitoring, conflict-resolution protocols, and human-in-the-loop control.

4. **Development and deployment challenges**: deep planning is required for multi-agent integration. Best practices: phased rollouts, continuous monitoring, and simulation-based validation.

5. **Scalability issues**: increasing agent counts and task complexity introduce compute/network bottlenecks. Solutions: dynamic resource allocation, infrastructure optimization, scalable coordination algorithms.

6. **Security and privacy**: sensitive data traversing multiple agents increases risk. Measures: end-to-end encryption, strict access controls, and periodic security audits.

7. **Troubleshooting complexity**: emergent behaviors and dependencies complicate debugging. Tools: comprehensive logging, distributed tracing, unified monitoring dashboards.

8. **Ethics and bias**: data biases can be amplified in hypothesis generation and experimental design. Mitigation: bias audits, fairness-aware algorithms, and diverse training corpora.

### 3.2 DOMAIN-SPECIFIC LIMITATIONS IN SCIENCE

(Google Research Blog, Feb 19, 2025)

1. **Literature coverage**: cross-disciplinary or non-indexed publications may be missed. Remedy: widen retrieval modules to broader scientific corpora.

2. **Fact-checking**: proposals and hypotheses may contain factual errors. Remedy: integrate external verification tools and authoritative databases.

3. **Tool interoperability**: reliance on built-in tools limits domain-specific simulations. Remedy: API-driven plugins for scientific software.

4. **Evaluation misalignment**: Elo-like metrics can diverge from expert judgment. Remedy: hybrid evaluation combining automated scoring with domain-expert feedback.

5. **Limited expert validation**: early studies involved small expert cohorts and narrow targets. Remedy: larger-scale, cross-disciplinary expert evaluations to enhance generalization.

### 3.3 FUTURE WORK AND RESEARCH DIRECTIONS

1. **Modular collaboration architectures**: decompose the scientific pipeline into Generation, Reflection, Ranking, Evolution, Proximity, and Meta-review modules with encapsulation, transparency, and adaptivity; support tight human–AI collaboration.

2. **Fusion of generative AI and MAS**: integrate LLM components safely, strengthen coordination and distributed decision-making, ensure ethics and traceability, and optimize resource efficiency.

3. **Cognitive architectures and distributed MAS**: explore edge computing and network heterogeneity to improve robustness; investigate architectures such as Soar and ACT-R for commonsense reasoning and long-horizon goal management.

4. **External tools and fact verification**: expand cross-disciplinary retrieval; introduce fact-checking engines and authoritative knowledge bases; build plugin interfaces for scientific software.

5. **Evaluation and expert collaboration**: move beyond Elo-style metrics with hybrid frameworks; organize broader, more diverse expert reviews; implement Trusted Tester programs.

6. **Governance and community standards**: develop interoperability protocols, ethics and compliance norms, and reproducibility standards for multi-agent scientist systems; clarify legal accountability and funding mechanisms.

### CONCLUSION

By 2025, multi-agent AI scientist technologies have made significant progress across definitions and methods, representative systems, datasets and evaluation benchmarks, and real-world applications, while also clarifying key challenges and avenues for future research. Shared characteristics include agent specialization, pipeline-style workflow orchestration, closed-loop iteration, and minimal human intervention. Real-world cases spanning drug repurposing, vaccine design, and materials discovery demonstrate the potential of multi-agent sys-

tems to accelerate scientific discovery. Future work should focus on robustness, safety, ethical compliance, and scalable evaluation frameworks to enable sustainable transition from prototypes to large-scale deployment.

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
