# OpenReview forum: "InternAgent-DR: Advancing deep research with dynamic structured knowledge flow"
_ICLR.cc/2026/Conference — ICLR 2026 Conference Desk Rejected Submission_

### Official Review · Reviewer_sJkB · 2025-10-31

**Soundness:** 3
**Presentation:** 3
**Contribution:** 2
**Rating:** 2
**Confidence:** 4

**Summary:**

This paper introduces InternAgent-DR, a multi-agent deep-research framework that models scientific reasoning as a dynamic structured knowledge flow. Instead of relying on a linear task sequence, InternAgent-DR represents research workflows as directed acyclic graphs whose nodes correspond to subtasks such as search, solve, and answer, and whose edges encode knowledge dependencies. The system integrates three major components: a Knowledge Flow Planner that incrementally expands the research graph, a Knowledge Collector that executes outermost nodes through LLM-based agents equipped with tools, and a Knowledge Flow Refiner that dynamically modifies the graph based on intermediate results. This design enables both hierarchical decomposition and adaptive refinement of complex research tasks. Extensive experiments on GAIA, GPQA-diamond, HLE, and TRQA benchmarks demonstrate that InternAgent-DR achieves state-of-the-art performance, surpassing existing open- and closed-source deep-research systems such as OpenAI-DR, OWL, and Manus. Ablation studies confirm the effectiveness of structured planning and flow refinement, and case studies show interpretability and reproducibility advantages.

**Strengths:**

1. The paper proposes a conceptually original formulation of deep research as a dynamic, graph-based reasoning process.
2. The methodology is technically rigorous, with clearly formalized planner, collector, and refiner modules.
3. Empirical evaluations on GAIA, GPQA, HLE, and TRQA show strong and consistent performance improvements over state-of-the-art baselines.

**Weaknesses:**

1. The framework appears to be a recombination of existing ideas rather than a fundamentally new approach. The so-called “dynamic structured knowledge flow” closely parallels the workflow-graph paradigms already used in OWL, MiroFlow, and OpenAI Deep Research, where subtasks and dependencies are explicitly represented in similar ways. The three modules essentially re-label the standard planning–execution–reflection loop without introducing new reasoning mechanisms or learning algorithms. The contribution is primarily in system packaging and terminology, not in conceptual or methodological innovation.
2. The paper lacks theoretical grounding and algorithmic depth. There is no formal analysis of convergence, scalability, or computational complexity. Claimed adaptivity and generalization advantages remain qualitative.
3. The prompt design inside each agent plays a crucial role in determining performance, yet this factor is not systematically studied. Different baselines may use distinct prompting templates or instruction strategies, which can drastically affect outcomes. Without prompt-controlled ablations and prompt unification across baselines, it is unclear whether performance gains come from the proposed system design or from better prompt engineering.
4. The evaluation setting lacks fairness control. The authors compare to several proprietary systems but do not ensure identical resource budgets, prompt contexts, or tool access, making the comparison potentially biased.

**Questions:**

see weakness

---

> ### Author Response · Authors · 2025-11-23
> **Reply to Reviewer sJkB**
>
> We sincerely thank the reviewer for careful reading of our manuscript. Below, we address each point in detail.
> ## Weaknesses
> ### 1. No conceptual or methodological innovation.
> Thanks for your feedback. The reviewer mentioned OWL, Miroflow, and other previous deep research frameworks, all of which model problem-solving logic and execution plans based on sequential reasoning, as discussed in our manuscript. However, the fundamental innovation of InternAgent lies in its ability to model the agent’s internal problem-solving logic as a graph-based knowledge flow, shifting the reasoning process from a sequential structure into a dependency-based, structured knowledge flow. Throughout the paper, we emphasize that this **knowledge flow driven planning replaces sequential planning**, a point that seems to be overlooked by the reviewer. To accommodate this new modeling approach, we redesign the collaboration between the planner, collector, and refiner, as well as their internal reasoning operations, to adapt to flow reasoning logic. They are absolutely not a re-label. In summary, InternAgent-DR introduces a new conceptual approach to modeling an agent’s problem-solving logic, presenting a novel idea and framework, and implementing it systematically. This represents a conceptual or methodological innovation, rather than simply being a matter of system packaging and terminology.
> ### 2.  lacks of theoretical grounding and algorithmic depth.
> Although analyses of convergence and computational complexity are not commonly found in deep research studies, we have still attempted to provide an analysis for the reviewer here.
>
> **Convergence**: We believe that the convergence of a deep research agent should be defined as its ability to return a response to the user within a finite amount of time. We control the system's convergence by limiting the number of nodes in the knowledge graph, ensuring that the system can always respond to a user query within a given time, or provide an answer to "I don't know."
>
> **Scalability, adaptivity and generalization**: We believe that the strong performance across multiple agentic and scientific benchmarks is sufficient to demonstrate that our approach possesses high scalability, generalization ability, and adaptability.
>
> **Computational complexity**: Although computational complexity is typically not systematically analyzed in DR agents, we still aim to provide the reviewer with a relevant analysis.
>
>   **Linear Planner**: Executes n subtasks sequentially, with the critical path length being n. The wall-clock time is approximately:
>   $$  Tseq≈n⋅tv$$
>
>   **Graph Planner (Knowledge flow Parallelism)**: Modeled as a dependency graph, where nodes and edges can be evaluated in parallel, resulting in higher total computational cost:
>   $$  Wgraph≈n⋅tv+E⋅te$$
>   However, with parallel execution, the wall-clock time is approximately:
>   $$  Tgraph,parallel≈n⋅tvpv+E⋅tepe+d⋅tsync$$
>   where $$d≪n$$ represents the depth of the graph.
>   Conclusion:
>  For the same number of subtasks, the Flow Planner significantly reduces the wall-clock time through parallel execution of subtasks and a shorter critical path ( $$d≪nd$$ ).
> ### 3. The impact of prompt design has not been sufficiently explored.
> For the agent workflow community, prompts, agents, and the internal roles of the system are closely interconnected, making it a system-level design issue. Different prompts can lead to significant performance variations across different systems, which means there is no universal baseline prompt or concept of a "superior" prompt. Consequently, it is uncommon to perform ablation studies that replace prompts within a system. This is a widely recognized and accepted research phenomenon within the community[1,2]. However, we have updated the prompt-related information in the appendix to clarify the prompts.
>
> [1] Tongyi DeepResearch Team, Li B., Zhang B., Zhang D., Huang F., Li G., Chen G., Yin H., Wu J., Zhou J., et al. (2025). Tongyi DeepResearch Technical Report. arXiv preprint arXiv:2510.24701.
>
> [2] Tian, H., Wang, C., Yang, B., Zhang, L., & Liu, Y. (2025). A Taxonomy of Prompt Defects in LLM Systems. arXiv preprint arXiv:2509.14404.
>
> ### 4. The evaluation setting lacks fairness control.
> We have presented the base models of different baseline frameworks in the main table. To ensure fairness, all other methods compared are based on their reported results, which always utilize base models more powerful than ours. Additionally, we present the performance of OWL after aligning it with our weaker base model, as well as the results of our system using flagship models on the GAIA test set, highlighting the strong robustness of our InternAgent-DR.
> As for the tools, we have listed all the tools we used in the appendix, which are largely aligned with those used in other agent frameworks.

---

### Official Review · Reviewer_hqXd · 2025-10-31

**Soundness:** 3
**Presentation:** 3
**Contribution:** 3
**Rating:** 6
**Confidence:** 4

**Summary:**

This paper introduces InternAgent-DR, a multi-agent system for complex scientific reasoning and problem-solving. It models research as a dynamic structured knowledge flow, where nodes represent subtasks and edges encode dependencies, enabling adaptive planning, reasoning, and refinement. The framework integrates three modules—Knowledge Flow Planner, Knowledge Collector, and Flow Refiner—to iteratively expand, execute, and adjust research plans. Experiments on benchmarks such as GAIA, GPQA, HLE, and TRQA show state-of-the-art performance, suggesting improved adaptability and reasoning depth compared to both single-agent and static multi-agent systems

**Strengths:**

- Representing a deep research task as a graph is interpretable and flexible - and it avoids common issues with single-/multi-agent issues about serial plan execution
- The pretty comprehensive evaluation benchmark results demonstrate great performance improvements

**Weaknesses:**

- Issues for fair comparison in evaluation: for an open-sourced agentic framework, I'd expect the authors to use the same model OWL/MiroFlow used to compare with them. We can't infer from the figure directly which model these open-source baselines used
- Limited information about prompting: It is unclear how the system works in detail from reading this paper, since no exact prompts for each stage were included. It will be helpful if the authors can share the prompts for each component in the appendix.
- The InternPlanner is trained via distillation on ~10k examples derived from Wikipedia-based or synthetic graphs, yet the paper offers little evidence of out-of-distribution robustness. Without evaluation on unseen domains or unseen question distributions, it is unclear whether the planner generalizes beyond GAIA-like setups.
- Also, it is unclear to me whether *training the planner* is strictly necessary -- my guess is that it could work out of the box by prompting with a stronger model. I think some experiments for "prompting+strong base model" would help justify the need for a trained planner

**Questions:**

- How is the InternPlanner’s generalization evaluated beyond its distillation source domain? Have you tested it on novel task types or unseen topics?
- Can you share more details about the prompts for each components
- I'd love to see an ablation justifying the need of "training the planner", eg replace the trained planner by simplying prompting a stronger LLM and measure the end performance

---

> ### Author Response · Authors · 2025-11-23
> **Reply to Reviewer hqXd**
>
> We sincerely thank the reviewer for careful reading of our manuscript. Below, we address each point in detail.
> ## Weaknesses
> ### 1. Issues for fair comparison in evaluation.
> Thanks for your suggestions. We have updated the table to include the base models used by various open-source frameworks. As shown in the table, while most open-source baselines rely on flagship models, we have chosen to use o4-mini due to cost considerations and its faster response times. Despite this, InternAgent still demonstrates competitive performance comparable to methods using flagship models. Additionally, we test with the Gemini2.5 pro model on GAIA, and the results show that when applying a stronger base model, our InternAgent-DR demonstrates a significantly greater performance advantage.
> | Method          | Base Model       | Level1 | Level2 | Level3 | Avg  |
> |-----------------|------------------|--------|--------|--------|------|
> | OWL             | o4-mini          | 67.9   | 55.8   | 23.1   | 55.8 |
> | OWL             | Gemini 2.5 Pro   | 84.9   | 68.6   | 42.3   | 69.7 |
> | InternAgent-DR  | o4-mini          | 90.6   | 76.7   | 50.0   | 76.9 |
> | InternAgent-DR  | Gemini 2.5 Pro   | **92.5** | **80.2** | **53.8** | **80.0** |
> ### 2. Limited information about prompting.
> Thanks for your suggestion, we have updated the appendix to include the prompts, and all codes will be released.
> ### 3. Generalization of InternPlanner.
> Thank you for your suggestion. Firstly, the scientific questions sampled for training InternPlanner do not present a domain gap with any of the test benchmarks. All training data were deduplicated before inclusion to ensure no domain overlap. To demonstrate the generalization capability of InternPlanner, we also conducted tests on the GPQA datasets, and the results will be added to Table 3. As shown, InternPlanner achieves performance improvements on both GAIA and GPQA, demonstrating the strong generalization capability of its training.
> | Planner                 | Level1 | Level2 | Level3 | Avg   | Bio   | Chem  | Phys  | Avg.  |
> |-------------------------|--------|--------|--------|-------|-------|-------|-------|-------|
> | **Qwen3-8B**            | 58.49  | 46.51  | 11.54  | 44.85 | 52.63 | 56.99 | 75.58 | 64.65 |
> | **InternPlanner-8B (ours)** | 70.25  | 67.44  | 34.61  | 66.06 | 63.16 | 60.22 | 86.05 | 71.64 |
> | **Qwen3-32B**           | 77.36  | 67.44  | 30.77  | 64.81 | 68.42 | 59.14 | 84.88 | 71.15 |
> | **InternPlanner-32B**   | 84.91  | 70.93  | 42.31  | 70.91 | 78.95 | 67.74 | 94.19 | 80.24 |
>
> ### 4. The necessity of training InternPlanner.
> You’re absolutely right, and we appreciate your observation. We apologize for any confusion caused. The results in the main table are not based on the use of InternPlanner, but rather on the o4-mini base model, just like you mentioned, which combines a strong base model with prompt configuration. The training of InternPlanner is actually an experiment we're exploring for future model training. Our current results suggest that effective training can improve open-source models for our workflow. Going forward, we plan to explore replacing o4-mini with our trained models. We're excited about these future developments, and we’ll keep you updated.
> ## Questions
> ### 1. Generalization of InternPlanner.
> As discussed in weakness 3, our training is not confined to the GAIA domain. Additional test results demonstrate that InternPlanner exhibits strong generalization capabilities across multiple tasks.
> ### 2. Limited information about prompting
> We'll update the appendix to include the promps.
> ### 3. An ablation justifying the need of "training the planner"
> Your observation is correct that the shown performance is based on o4-mini. The training of InternPlanner is currently an initial attempt as part of our future plans for model training.

---

### Official Review · Reviewer_mnNE · 2025-11-01

**Soundness:** 3
**Presentation:** 3
**Contribution:** 2
**Rating:** 4
**Confidence:** 4

**Summary:**

This paper proposes InternAgent-DR, a deep-research system that constructs and evolves a dynamic structured knowledge flow. Instead of linear task pipelines, the method builds a DAG-structured research graph to explicitly model subproblem dependencies, support parallel exploration, and adapt structure during execution. The system includes (i) a flow planner, (ii) a knowledge collector with tool-augmented LLM agents, and (iii) a flow refiner for graph-level self-revision. Experiments on GAIA, GPQA, HLE, and TRQA show state-of-the-art or competitive performance. Ablations indicate benefits from both structured planning and dynamic refinement.

**Strengths:**

1. Introduces a dynamic DAG-based research flow that evolves during execution, distinct from fixed linear pipelines. Emphasizes reflective graph refinement, paralleling execution and iterative plan refinement.

2. Strong empirical results across diverse and challenging benchmarks.

3. Good ablations isolating planner and refiner effects, trained planners outperform base models.

4. Case studies demonstrate interpretability and node-level execution traceability.

5. Presents a coherent and well-structured formulation of the system mechanism and workflow design.

**Weaknesses:**

1. **Methodology and Limitations**
- The method lacks theoretical grounding. It is unclear why the knowledge-flow refinement should systematically improve deep-research trajectories, and what precisely is being optimized during refinement (structure? retrieval quality? evidence routing?). The contribution currently rests largely on empirical results without a principled justification.

- The core idea of dynamic plan adjustment based on execution status is compelling. However, the paper fails to discuss or compare its approach with other methods that employ similar reflective mechanisms. For instance, Aworld utilize "guard reasoning" for dynamic reflection and plan expansion, which might be a more cost-effective approach. While the proposed method may achieve higher accuracy, a comparative discussion is necessary to position its contribution accurately.

- The paper lacks a discussion of the method's potential limitations and failure scenarios.
For example, does frequent plan refinement introduce prohibitive computational overhead? On challenging problems, could the agent get trapped in refinement loops or even diverge from a correct solution? If an execution error is caused by a faulty tool call, could the subsequent plan adjustments lead the agent further astray?

- A clear analysis of failure cases and the system's fault tolerance mechanisms is currently missing.


2. **Analysis of the Refiner**

- The ablation study identifies the Refiner as the source of the largest performance gains, yet the analysis of its effectiveness is not enough. the analysis of its effectiveness is insufficient and does not provide a deep dive into its mechanism.

- Moreover, I am curious about the performance improvement on GAIA Level-1 tasks, as they typically have clearer execution paths. As practices from other agents like Skywork-DR suggest, a robust planner should handle such tasks effectively without extensive refinement. This raises serious questions about the baseline Flow Planner used in the study; its design may be suboptimal, thereby inflating the perceived contribution of the Refiner.  Stronger planner baselines or cross-agent planner comparisons are required to isolate the true contribution.

3. **Collector–Refiner Interface Not Specified**

The paper does not clearly describe what form the collected information is provided to the Refiner in(e.g., raw tool traces, distilled node summaries,or full textual context). Without clarity on representation and conditioning, it is difficult to understand how the Refiner:

- incorporates execution signals,

- avoids being misled by noisy tool outputs, and maintains reasoning fidelity during refinement.

Greater transparency here would improve interpretability and reproducibility.

4. **Knowledge Collector Reliability**

Execution correctness in deep-research settings is not binary. Search tools may “succeed” but return irrelevant text; browser automation may execute actions that do not advance reasoning. The paper does not clarify:

- How execution quality is evaluated beyond binary success/failure
- How noisy or irrelevant evidence is suppressed
- How incorrect tool outputs are prevented from influencing refinement

The paper lacks theoretical innovation and is primarily an engineering-driven empirical system contribution.

**Questions:**

1. How does the Refiner theoretically improve reasoning? What objective or latent structure is being optimized?

2. Why does refinement boost GAIA-Level-1 task, does this imply a weak baseline planner?

3. How do you prevent refinement loops, over-expansion, or divergence under noisy tool execution?

4. How are node execution signals summarized and fed into the Refiner? Are there safeguards to identify knowledge quality before refinement?

5. Can you add comparisons with AWorld / Skywork-DR and clarify base model settings for all agent baselines?

6. What is the compute overhead per refinement cycle, and is there a refinement-budget policy?

---

> ### Author Response · Authors · 2025-11-23
> **Reply to Reviewer mnNE (Part 1)**
>
> We sincerely thank the reviewer for careful reading of our manuscript. Below, we address each point in detail.
> ## Weaknesses
> ### 1. Methodology and Limitations
> The reviewer misconstrues our core contribution. Our central innovation is **not** “refinement,” but the **transformation of sequential planning logic into a DAG-structured Knowledge Flow**. This yields two fundamental insights:
>
>  (1) the knowledge flow explicitly encodes subtask dependencies, enabling precise and minimal context selection;
>  (2) this structured logic makes genuinely **dynamic, structure-aware refinement** possible—far beyond what sequential planners can express.
>
> As detailed in the paper, our entire system (including reflection) is built on this modeling paradigm. Consequently, our refinement mechanism is **conceptually distinct** from prior sequential-plan “reflection”: while sequential agents can only insert/delete steps along a linear chain, our DAG-based refiner can **modify local flows, adjust dependency edges, and introduce new search nodes**, expanding the decision space. The effectiveness of both the planning strategy and dynamic refinement is validated in our ablations.
>
> Regarding potential failure modes, the **refiner itself serves as the fault-tolerance mechanism**: when a subnode fails, it replans the affected subgraph. To prevent unbounded exploration, we cap the number of nodes and terminate gracefully when resolution is impossible. More fine-grained strategies will be incorporated in future versions.
> ### 2. Analysis of the Refiner
> We agree with the reviewer that Level-1 tasks often follow relatively fixed execution paths. However, our core contribution is **not** to improve planner capacity, but to introduce a fundamentally stronger **agent reasoning and planning logic**, and to design a system that fully exploits this logic to achieve better results. Moreover, in an open-network environment, execution failures are unavoidable. In such cases, the refiner provides essential fault tolerance by replanning local subgraphs and mitigating the impact of subtask failures—an especially important property given the limited capacity of weak planners and the instability of real-world tool environments. Meanwhile, for complex tasks that cannot be resolved by a static graph alone, InternAgent-DR also substantially improves multi-step reasoning performance.
>
> Crucially, the reviewer’s claim that “the refiner is the main source of performance gains” is inaccurate. As emphasized in the Methodology and Limitations section, our core contribution is **the transformation of sequential execution logic into a DAG-structured Knowledge Flow and further development upon the insight**. Both the flow planner and the refiner were co-designed around this modeling paradigm and cannot be meaningfully isolated. Prior agent systems are fundamentally built around sequential plans and thus cannot support our DAG-based refiner; conversely, our planner cannot operate within their architectures. These are system-level structural dependencies, not modular components that can be independently ablated.
>
> Besides, our ablations show that under identical backbones, **the flow planner significantly outperforms sequential planners**, addressing concerns regarding suboptimality. Furthermore, when using a stronger planner model, the refiner still yields substantial gains—but **only within the dynamic Knowledge Flow framework**. Any discussion detached from this core insight is therefore not meaningful.
> | Sequential Planner | Flow Planner | Refiner | Level 1 | Level 2 | Level 3 | Avg   | Bio   | Chem  | Phys  | Avg.  |
> |--------------------|--------------|---------|---------|---------|---------|-------|-------|-------|-------|-------|
> | ✓                  | --           | --      | 67.92   | 55.81   | 23.07   | 55.76 | 57.89 | 54.84 | 88.37 | 71.21 |
> | --                 | ✓            | --      | 73.58   | 63.95   | 30.77   | 61.82 | 57.89 | 59.14 | 89.53 | 73.74 |
> | --                 | ✓            | ✓       | **90.56** | **76.74** | **50.00** | **76.96** | **84.21** | **79.57** | **96.51** | **87.37** |
>
> ### 3. Collector–Refiner Interface Not Specified
> We have updated the appendix to include the prompt specifications for each agent component, making the design of each module more transparent. All collected information is summarized and stored within the corresponding nodes. More detailed implementation will be available in our upcoming code release.

---

> > ### Author Response · Authors · 2025-11-23
> > **Reply to Reviewer mnNE (Part 2)**
> >
> > ### 4. Knowledge Collector Reliability
> > The Collector is solely responsible for information gathering. We followed common practice by integrating a set of common-used tools without applying any special design modifications. In contrast to sequential-plan-based approaches, which can only pass outputs linearly from one layer to the next, our Knowledge Flow enables the refiner to assess the reliability of task results according to the logical dependencies of knowledge flow. It can determine whether the gathered information contributes to solving the overall problem, and mitigate potential errors by deleting nodes, adjusting dependency edges, or adding verification queries for retrieved results. The reliability of individual tools is not the focus of our current analysis; we plan to explore and optimize this aspect in future work.
> >
> > ## Questions
> > ### 1. How does the Refiner theoretically improve reasoning? What objective or latent structure is being optimized?
> > The improvements in reasoning are not achieved by the refiner itself, but rather stem from the structured modeling provided by the Knowledge Flow. The refiner primarily contributes fault tolerance and can dynamically adjust the logical structure based on collected information, further enhancing overall performance. Its sole optimization objective is to **construct a comprehensive knowledge flow to solve the final task.**
> > ### 2. Why does refinement boost GAIA-Level-1 task, does this imply a weak baseline planner?
> > First, our flow planner demonstrates overall **stronger** performance compared with a sequential planner as shown in Table 2. Second, the refiner enhances robustness by replanning subtask nodes in response to failures caused by network variability or other noise, thereby improving performance on GAIA Level-1 tasks.
> > ### 3. How do you prevent refinement loops, over-expansion, or divergence under noisy tool execution?
> > We impose a maximum limit on the number of nodes. If the task cannot be completed within this limit, InternAgent determines whether an answer can be produced based on the information collected so far; if not, it responds with “unknown.” More detailed prevention mechanisms will be explored in future work.
> > ### 4. How are node execution signals summarized and fed into the Refiner? Are there safeguards to identify knowledge quality before refinement?
> > Our refiner prompt has been included in the appendix. Currently, we do not implement a dedicated knowledge quality filtering system as common practice; the refiner alone evaluates the completeness of the Knowledge Flow and the likelihood of solving the task to make corrections to the overall flow. We will explore more sophisticated mechanisms for this in future work.
> > ### 5. Can you add comparisons with AWorld / Skywork-DR and clarify base model settings for all agent baselines?
> > We have included comparisons with AWorld and Skywork-DR in Table 1, and clarify the information regarding the agent base models that we were able to obtain.
> > ### 6. What is the compute overhead per refinement cycle, and is there a refinement-budget policy?
> > We make a resource comparison in the table below. Thanks to InternAgent-DR's more cost-effective and efficient base model, we achieve advantages in both cost and response time. Currently, we only impose a maximum node limit to prevent excessive resource usage. In future work, we will explore more sophisticated refinement-budget policies.
> > | Method          | API Cost ($/query) | Time (s/task) |
> > |-----------------|---------------------|----------------|
> > | InternAgent-DR  | 0.86                | 177            |
> > | OWL             | 1.20                | 236            |

---

### Official Review · Reviewer_FiJa · 2025-11-01

**Soundness:** 2
**Presentation:** 3
**Contribution:** 2
**Rating:** 2
**Confidence:** 4

**Summary:**

The paper proposes InternAgent-DR **,** a multi-agent deep research system that builds and continually refines a knowledge flow (planner → collector → refiner) to coordinate subtasks and dependencies. Experiments on GAIA, HLE, GPQA, and TRQA report strong or SOTA results, with ablations showing benefits from structured planning and dynamic refinement.

**Strengths:**

* Clear, principled formulation of a dynamic knowledge DAG.The paper formalizes nodes (typed subtasks with state/context) and edges (dependency relations) and argues this captures non-linear dependencies while enabling parallelism and verifiable execution; the end-to-end loop (planner/collector/refiner) is well-motivated.
* Compelling empirical results across diverse benchmarks. InternAgent-DR (especially with o4-mini) achieves leading scores on GAIA and HLE and strong performance on GPQA/TRQA; ablations isolate gains from the flow planner and  flow refiner .
* Reasonably transparent system details and tooling. The paper lists concrete tool wrappers used by the Knowledge Collector and describes a two-mode summarizer for QA vs. report generation—useful for reproducibility and for readers attempting to re-implement.

**Weaknesses:**

## Major Weaknesses

1. **Fragmented main results and missing agentic baselines.** Table 1 shows many unreported entries (“–”) across GAIA/GPQA-diamond/HLE, resulting in fragmented and unfair evaluations. Notably, several agentic frameworks lack GPQA-diamond results in the table (e.g., OpenAI-DR has GAIA/HLE but no GPQA), so comparisons on this science-heavy benchmark fall back to base models rather than agentic systems. This weakens claims about advantages over agents on GPQA-diamond.
2. **Baseline pool under-represents the field and conflates categories.**
   The comparison set mixes a few proprietary/open frameworks with many base models, but omits a broad slate of contemporary DR agents. Please distinguish and cover both categories: (a) Open-source DR models: WebDancer, WebSailor, DeepResearcher, WebShaper etc. The authors mentioned WebDancer in related work but did not evaluate it. (b) Open-source DR frameworks : JoyAgent, OAgents, WebWeaver, DeerFlow, AFlow, Skywork etc.
3. **Limited methodological novelty beyond well-known agent paradigms**. The core pipeline—Planner → Collector → Refiner running over a node-edge knowledge flow/DAG —largely mirrors prevalent graph-orchestrated agent patterns (orchestrator/worker, iterative refinement) rather than introducing a fundamentally new algorithmic idea. The paper formalizes incremental graph expansion and a set of graph edits (Add/Remove/Modify node/edge) but these are standard graph transforms; there is no new planning/search objective, scheduler, or learning signal that makes the “knowledge workflow” qualitatively different from prior graph-based agent stacks (e.g., LangGraph-style DAG orchestration or ToT/GoT/AoT-like task decomposition). This makes the contribution feel engineering-centric rather than conceptually novel.
4. **Attribution of gains is underspecified, ablations are narrow.** The ablation isolates “structured planning” and “refinement,” but remains confined to GAIA/GPQA and does not disentangle improvements due to (a) planner prompting/training, (b) tool repertoire, or (c) execution policy (parallelism/scheduling). Table 2 and Table 3 show benefits from the Flow Planner/Refiner and from the fine-tuned InternPlanner, but there is no head-to-head against graph-driven agents with matched backbones/tool budgets to attribute gains to the proposed workflow rather than to planner quality or base model capacity. A stronger attribution study is needed (e.g., same backbone, same tools, replacing only the orchestration logic).

## Minor Weaknesses

1. **Reporting gaps on cost, reliability, and failure modes.** There is little quantification of compute/tool-usage cost, latency, or robustness (e.g., tool failure rates, noisy retrieval) under the dynamic DAG loop, and no human evaluation of usability or interpretability for researchers
2. **Claims of general superiority outpace evidence; scheduling/efficiency untested.** The text asserts that a general-purpose, knowledge-flow agent “outperforms specialized systems” (e.g., on HLE), and repeatedly describes parallel execution and dynamic refinement as efficiency enablers, but provides no quantitative analysis of scheduler behavior (throughput/latency), failure recovery costs, or scaling vs. graph size. Without such evidence (speedups, queueing, deadlock avoidance, token/compute budgets per node type), the efficiency narrative remains claim-level rather than experimentally supported.
3. **Dataset and generalization concerns**. The InternPlanneris distilled from o4-mini on synthetic Wikipedia-derived graphs; the paper provides limited analysis of transfer to open-domain research beyond the chosen benchmarks and limited discussion of potential training-data leakage/contamination.
4. **Lack of significance analysis**. Due to network/tool latency, API nondeterminism (e.g., retrieval variance, page dynamics), and sampling randomness of the backbone LLMs, DR agents typically exhibit high run-to-run variance. Their experiments report single-point scores without confidence intervals or multi-seed statistics; no paired significance tests are provided to establish that the reported gains are robust rather than noise.

**Questions:**

The appendices list tool wrappers and the two-mode summarizer, and the planner dataset (10k) is described, but the paper does not clearly commit to releasing code, prompts, or the planner dataset, limiting external verification and community adoption. A precise release plan (what/when) would materially strengthen the paper.

---

> ### Author Response · Authors · 2025-11-23
> **Reply to Reviewer FiJa (Part 1)**
>
> We sincerely thank the reviewer for careful reading of our manuscript. Below, we address each point in detail.
> ## Major weaknesses
> ### 1. Fragmented main results and missing agentic baselines.
> We thank the reviewer for the valuable feedback. We have updated Table 1 to make the evaluation more complete and fair. Specifically, we have added the previously missing results of testable agent frameworks on the GPQA-diamond benchmark, based on their publicly reported or reproducible performance. For methods that cannot be reproduced or systematically evaluated due to cost constraints or lack of public availability (e.g., OpenAI-DR), we are unable to provide the corresponding results.
> ### 2. Baseline pool under-represents the field and conflates categories.
> We have added results for additional baseline methods (such as the Web series) to ensure a more comprehensive evaluation. As in the first point of the major weaknesses, some of these methods cannot be evaluated on HLE or GPQA because running each agent system on these datasets incurs costs of thousands of dollars. Moreover, so many methods mentioned by the reviewers are published after our submission, and their performance is generally lower than ours, further demonstrating the strong competitiveness of InternAgent-DR.
> ### 3. Limited methodological novelty beyond well-known agent paradigms.
> We appreciate the reviewer’s comments. The graph-based agent stacks mentioned by the reviewer (e.g., LangGraph-style frameworks) primarily use graph structures to schedule collaboration among multiple agents—that is, to represent workflows or interaction patterns between agents. This is conceptually different from our dynamic knowledge flow: such frameworks focus on coordination across multiple agents, whereas our work focuses on modeling the internal reasoning logic and execution planning within agent systems.
>
> Specifically, existing frameworks use graphs to depict inter-agent collaboration. Their internal problem-solving logic typically still follows a **sequential** reasoning process, consistent with our discussion in the paper. In contrast, **the core innovation of InternAgent-DR lies in the first deep research agent framework modeling the internal problem-solving process as a DAG-like dynamic knowledge flow**, transforming reasoning from a sequential chain into a dependency-aware structured knowledge flow.
>
> To support this new reasoning paradigm, we redesigned the collaboration mechanisms among the planner, collector, and refiner, and adapted their internal cognitive operations accordingly. In summary, InternAgent-DR is built upon a novel insight of conceptualizing the problem-solving logic inside an agent. It introduces a novel conceptual framework and provides a systematic implementation, achieving both fundamental algorithmic innovations and system-level integration and optimization.

---

> > ### Author Response · Authors · 2025-11-23
> > **Reply to Reviewer FiJa (Part 2)**
> >
> > ### 4. Attribution of gains is underspecified, ablations are narrow.
> > In Table below, we evaluate three configurations: (1) a sequential planner versus our knowledge-flow planner, and (2) the knowledge-flow planner with and without the refiner. Under identical backbones and toolsets, the comparison between the sequential and flow planners clearly demonstrates the superiority of our flow-based logical modeling. The refiner further yields consistent and measurable gains. The results are reported on GAIA and GPQA.
> > | Sequential Planner | Flow Planner | Refiner | Level 1 | Level 2 | Level 3 | Avg   | Bio   | Chem  | Phys  | Avg.  |
> > |--------------------|--------------|---------|---------|---------|---------|-------|-------|-------|-------|-------|
> > | ✓                  | --           | --      | 67.92   | 55.81   | 23.07   | 55.76 | 57.89 | 54.84 | 88.37 | 71.21 |
> > | --                 | ✓            | --      | 73.58   | 63.95   | 30.77   | 61.82 | 57.89 | 59.14 | 89.53 | 73.74 |
> > | --                 | ✓            | ✓       | **90.56** | **76.74** | **50.00** | **76.96** | **84.21** | **79.57** | **96.51** | **87.37** |
> >
> > All experiments adopt a fixed toolsets, which is a common practice in agent systems, and these components are not central to our methodological contribution; thus, we do not ablate them.
> > Regarding comparisons with other graph-driven agents, we believe there might be a conceptual misunderstanding, which we clarified in Major weakness 3. Furthermore, Table 1 have been updated to annotate the backbone models, many of which rely on substantially more expensive flagship models with longer context windows (e.g., Claude 3.5/Gemini 2.5 pro etc.), whereas our system consistently uses o4-mini. To demonstrate that the o4-mini is our cost-effective chosen model rather than a performance-tailored choice, we additionally report results where OWL and InternAgent-DR are run with the same o4-mini and Gemini 2.5 Pro backbone. Even with less powerful backbone o4-mini, InternAgent-DR outperforms other frameworks, underscoring the advantage of InternAgent-DR and its dynamic knowledge flow. Toolsets and prompts are integral to each agent system’s design, and for fairness, we do not modify them for competing methods.
> > | Method          | Base Model       | Level1 | Level2 | Level3 | Avg  |
> > |-----------------|------------------|--------|--------|--------|------|
> > | OWL             | o4-mini          | 67.9   | 55.8   | 23.1   | 55.8 |
> > | OWL             | Gemini 2.5 Pro   | 84.9   | 68.6   | 42.3   | 69.7 |
> > | InternAgent-DR  | o4-mini          | 90.6   | 76.7   | 50.0   | 76.9 |
> > | InternAgent-DR  | Gemini 2.5 Pro   | **92.5** | **80.2** | **53.8** | **80.0** |
> >
> > ## Minor Weaknesses
> > ### 1. Reporting gaps on cost, reliability, and failure modes.
> > In the table below, we report the invocation cost and response latency of InternAgent-DR and OWL. Benefiting from a more lightweight and cost-efficient foundation model o4-mini, as well as a more streamlined reasoning chain, InternAgent-DR achieves consistently lower cost and latency compared with other performance-comaparible frameworks.
> >
> >
> > It is worth noting that tool failures and noisy retrieval are external, non-algorithmic factors. All evaluations were conducted under real network conditions, where typical tool failure rates and retrieval noise naturally occur. As a result, the reported performance directly reflects the robustness of our system in complex real-world scenarios rather than in idealized settings.
> > ### 2. Claims of general superiority outpace evidence; scheduling/efficiency untested.
> > To demonstrate the generality of our approach, we evaluated InternAgent-DR not only on the general benchmark GAIA but also across three distinct scientific benchmarks. The results consistently show that InternAgent-DR maintains robust performance across diverse scenarios.
> > | Method          | API Price ($/query) | Time (s/task) |
> > |-----------------|----------------------|----------------|
> > | InternAgent-DR  | 0.86                 | 177            |
> > | OWL             | 1.20                 | 236            |
> >
> > It is important to emphasize that our primary contribution lies in enhancing system performance through graph-structured logical modeling. Mentions of efficiency appear only twice in non-critical parts of the paper and do not constitute a main claim. Parallel execution is solely employed to facilitate the efficient scheduling of the knowledge flow, rather than to optimize the efficiency of baseline methods. Therefore, the reviewer’s interpretation regarding efficiency reflects a misunderstanding of our work.

---

> > > ### Author Response · Authors · 2025-11-23
> > > **Reply to Reviewer FiJa (Part 3)**
> > >
> > > ### 3. Dataset and generalization concerns.
> > > The scientific questions sampled for training InternPlanner do not present a domain gap with all test benchmark. All training data were deduplicated before inclusion in training to ensure no domain overlap. Furthermore, all models used in the final experiments are **o4-mini**, and the entire framework operates in a fully **training-free** manner without any external knowledge bases, eliminating any possibility of data leakage.
> > > ### 4. Lack of significance analysis.
> > > We report the mean and variance over three runs on the GAIA benchmark.
> > > |      | Level1         | Level2        | Level3        | Avg          |
> > > |------|----------------|---------------|---------------|--------------|
> > > | InternAgent-DR | 89.8 (±2.9)  | 78.3 (±1.8)  | 53.8 (±3.8)  | 78.2 (±0.6) |
> > >
> > > The results indicate that while InternAgent-DR’s performance exhibits some fluctuation under network variability, the overall mean remains highly competitive.
> > > ## Questions
> > > We will release all codes and training data once upon acceptance of the paper, and will continue to provide updated, more powerful versions to facilitate reproducibility and further research in the community.

---

> > > > ### Comment · Reviewer_FiJa · 2025-11-24
> > > >
> > > > I appreciate the authors' detailed response and the additional efforts during the rebuttal. However, after carefully examining the updated manuscript and the rebuttal data, I find that the core issues regarding experimental rigor and baseline selection remain unresolved. Consequently, I will maintain my original score.
> > > >
> > > > My specific concerns are as follows:
> > > >
> > > > **1. Fragmented Results and Incomplete Evaluation (Table 1)**
> > > >
> > > > Despite the updates, **Table 1 remains highly fragmented, with significant missing entries (blank cells), particularly on the HLE and GPQA-diamond benchmarks.**
> > > >
> > > > * The reliance on "publicly reported" results for some baselines while leaving others blank prevents a comprehensive assessment.
> > > > * Without a unified evaluation on these challenging benchmarks, it is impossible to verify whether InternAgent-DR truly advances the state-of-the-art or simply occupies a different trade-off point. A rigorous study should prioritize running a consistent set of baselines on at least a representative subset of the data, rather than presenting a table full of gaps.
> > > >
> > > >  **2. Inappropriate Baseline Selection and Comparison**
> > > >
> > > > The new ablation studies and efficiency claims rely heavily on **OWL**, which is problematic for two reasons:
> > > >
> > > > * **Performance Comparison:** **The baseline pool lacks stronger, more relevant recent frameworks. Specifically, the paper fails to compare against** **MiroFlow**, which is highly relevant to the proposed flow-based approach. Omitting such critical baselines weakens the claim of superiority.
> > > > * **Efficiency Comparison:** **The efficiency advantage claimed over OWL is unconvincing. OWL is structurally known for high overhead due to its frequent** **nested retries**. Surpassing OWL in latency and cost does not inherently demonstrate that InternAgent-DR is efficient; it merely shows it is less redundant than a framework known for high consumption. A fair efficiency evaluation should be conducted against more streamlined agents, not just one with known bottlenecks.
> > > >
> > > > **3. Methodological Novelty**
> > > >
> > > > As noted in my initial review, the distinction between "inter-agent collaboration" and "internal knowledge flow" feels more like an engineering adaptation than a fundamental conceptual insight. While the implementation is functional, the novelty does not reach the threshold typically expected for ICLR.
> > > >
> > > > In summary, the author's response did not address the substantive issues in the paper, so I have decided to keep my score.

---

> ### Author Response · Authors · 2025-11-28
> **Concerns Regarding AI-Generated Reviews and Review Quality by Reviewer FiJa**
>
> Dear All:
>
> Thank you for taking the time to read our response. During the process of preparing our rebuttal, we have identified significant concerns regarding the quality and reliability of the reviews and response provided by **reviewers FiJA**. These issues are particularly troubling given that the reviewers have assigned a confidence level of 4 to the evaluation.
>
> Our concerns are further substantiated by an analysis conducted using a **third-party AI-generated text detection tool** (link: https://iclr.pangram.com/reviews?submission_number=830), which indicates that the review is highly likely to be **fully generated by AI**. Further, we would like to outline the specific issues we have observed:
>
> In the review
>
> 1. In Major Weakness 2, several cited methods were published after our submission, suggesting the list may have been automatically gathered by an AI system.
>
> 2. In Major Weakness 3, the reviewer conflates our knowledge-modeling graph with the execution graph used in LangGraph. This contradicts their own strengths section and may indicate conceptual confusion typical of AI hallucination.
>
> 3. Minor Weakness 1 requests human evaluation, which is not common practice in the Deep Research domain, remains an open question for the community, and resembles an AI-generated generic critique template.
>
> 4. In Minor Weakness 2, the reviewer claims we “repeatedly emphasize efficiency,” whereas efficiency is mentioned only twice in the entire paper, an inaccurate statement likely driven by AI hallucination.
>
> **As for the response from FiJa, the results from a third-party AI-generated text detection tool are provided here**: https://www.pangram.com/history/d9db34ac-d1bd-4dc4-99d4-072f37fbdd24?ucc=rxR7HeUgT8w. It shows that the response is also fully AI generated. We outline the issues we have observed:
>
> 1. The reviewer claimed that we did not make comparisons with MiroFlow, however, MiroFlow is clearly listed in the first row of our main table. This statement appears to be inaccurate and likely a result of AI hallucination.
>
> 2. The reviewer claims tha "Miroflow is highly relevant to the proposed flow-based approach", however, Miroflow is not relevant to knowledge flow at all. Miroflow0.1 follows a sequential planing multi agent framework while Miroflow 1.0 follows a react framework, none of which is flow or graph related. It's apprently an AI misubderstanding of the related name "flow" and such a misinterpretation would not be expected from a professional researcher.
>
> 3. The claim that “OWL is structurally known for high overhead” has never been made within the agent community. Neither the GitHub repository (camel-ai/owl) nor other discussions have indicated that OWL incurs high overhead, and OWL is a common used baseline framework by several previous works [1,2,3]. On the contrary, Table 7 in Efficient Agents [3] shows that OWL is a highly efficient baseline, running approximately 6× faster than several other frameworks. This statement from the reviewer is therefore not supported by any known evidence and is very likely hallucinated content generated by AI.
>
> 4. The reviewer’s criticism about “fragmented tables” and reliance on publicly reported results reflects a misunderstanding of common practice in the agent community. Re-running all baselines on new and computationally expensive benchmarks is prohibitively costly, so prior works including Tongyi Deep Research[4], MiroFlow/MiroThinker[5], FlashSearcher[6], routinely report only publicly reported baseline results. Thus, the claim that this reliance “simply occupies a different trade-off point” is unfounded, the reviewer’s remark is disproportionately demanding and misaligned with research common practice.
>
> [1] Zhu, H., Qin, T., Zhu, K., Huang, H., Guan, Y., Xia, J., ... & Zhou, W. Oagents: An empirical study of building effective agents, 2025. URL https://arxiv. org/abs/2506.15741.
>
> [2] Liu, Jiarun, et al. "JoyAgent-JDGenie: Technical Report on the GAIA." arXiv preprint arXiv:2510.00510 (2025).
>
> [3] Wang, N., Hu, X., Liu, P., Zhu, H., Hou, Y., Huang, H., ... & Zhou, W. (2025). Efficient agents: Building effective agents while reducing cost. arXiv preprint arXiv:2508.02694.
>
> [4] Team, T. D., Li, B., Zhang, B., Zhang, D., Huang, F., Li, G., ... & Jiang, Y. (2025). Tongyi DeepResearch Technical Report. arXiv preprint arXiv:2510.24701.
>
> [5] Team, M., Bai, S., Bing, L., Chen, C., Chen, G., Chen, Y., ... & Zhu, Z. (2025). MiroThinker: Pushing the Performance Boundaries of Open-Source Research Agents via Model, Context, and Interactive Scaling. arXiv preprint arXiv:2511.11793.
>
> [6] Qin, T., Chen, Q., Wang, S., Xing, H., Zhu, K., Zhu, H., ... & Zhou, W. (2025). Flash-searcher: Fast and effective web agents via dag-based parallel execution. arXiv preprint arXiv:2509.25301.
>
>
> These issues raise substantial concerns regarding the reliability and human authorship of the reviews.

---

### Author Response · Authors · 2025-12-01
**A consolidated summary of the reviewers’ comments for the Area Chair (Part 1)**

Dear Area Chair,

To facilitate your assessment, we provide a concise summary of how we have addressed the reviewers’ primary concerns.

**1. Missing agentic baselines and models open-source baselines used.**

The Reviewer FiJa, mnNE mentioned the missing baselines while Reviewer sJkB and hqXd concerned about the model baselines for fainess. **We have updated Table 1 to make the evaluation more complete.** Specifically, we have added the previously missing results of testable agent frameworks on the GPQA-diamond benchmark, based on their publicly reported or reproducible performance. For methods that cannot be reproduced or systematically evaluated due to cost constraints (The evaluation of HLE and GPQA of an agent workflow is pretty costly) or lack of public availability (e.g., OpenAI-DR), we are unable to provide the corresponding results. Moreover, so many methods mentioned by the reviewers are published after our submission, and their performance is generally lower than ours, further demonstrating the strong competitiveness of InternAgent-DR. **And we also add the base models of the baselines used to show fairness**, while most open-source baselines rely on flagship models. We have chosen to use the o4-mini due to cost considerations and its faster response times. Despite this, InternAgent still demonstrates competitive performance comparable to methods using flagship models. Additionally, we test with the Gemini2.5 pro model on GAIA, and the results show that **when applying a stronger base model, our InternAgent-DR demonstrates a significantly greater performance advantage.**

| Method          | Base Model       | Level1 | Level2 | Level3 | Avg  |
|-----------------|------------------|--------|--------|--------|------|
| OWL             | o4-mini          | 67.9   | 55.8   | 23.1   | 55.8 |
| OWL             | Gemini 2.5 Pro   | 84.9   | 68.6   | 42.3   | 69.7 |
| InternAgent-DR  | o4-mini          | 90.6   | 76.7   | 50.0   | 76.9 |
| InternAgent-DR  | Gemini 2.5 Pro   | **92.5** | **80.2** | **53.8** | **80.0** |

**2. Limited novelty and algorithmic depth.**

The reviewers FiJa and sJkB attempt to challenge our novelty by conflating our innovation at the level of planning logic with the graph-based collaboration used in conventional agents, while reviewer mnNE has also misunderstood our core contribution. The graph-based agent stacks mentioned by the reviewer (e.g., LangGraph-style frameworks) primarily use graph structures to schedule collaboration among multiple agents, whereas our work focuses on modeling the internal reasoning logic and execution planning within each agent in the system.  Our central innovation is **the transformation of sequential planning logic into a DAG-structured Knowledge Flow. **This yields two fundamental insights:

(1) the knowledge flow explicitly encodes subtasks dependencies, enabling precise and minimal context selection;

(2) this structured logic makes locally, dynamic, structure-aware refinement of the execution plan possible—far beyond what sequential planners can express.

As detailed in the paper, our entire system (including reflection) is built on this modeling paradigm. Consequently, our refinement mechanism is conceptually distinct from prior sequential-plan “reflection”: while sequential agents can only insert/delete steps along a linear chain, our flow-based refiner can **modify local flows, adjust dependency edges, and introduce new search nodes, expanding the decision space**. The effectiveness of both the planning strategy and dynamic refinement is validated in our ablations.

| Sequential Planner | Flow Planner | Refiner | Level 1 | Level 2 | Level 3 | Avg   | Bio   | Chem  | Phys  | Avg.  |
|--------------------|--------------|---------|---------|---------|---------|-------|-------|-------|-------|-------|
| ✓                  | --           | --      | 67.92   | 55.81   | 23.07   | 55.76 | 57.89 | 54.84 | 88.37 | 71.21 |
| --                 | ✓            | --      | 73.58   | 63.95   | 30.77   | 61.82 | 57.89 | 59.14 | 89.53 | 73.74 |
| --                 | ✓            | ✓       | **90.56** | **76.74** | **50.00** | **76.96** | **84.21** | **79.57** | **96.51** | **87.37** |

**3. Limited information about prompting.**

The reviewers mnNE, hqXd and sJkB mentioned that we provide limited information about prompting. We have updated the appendix to include the prompts, and all codes will be released.

---

> ### Author Response · Authors · 2025-12-01
> **A consolidated summary of the reviewers’ comments for the Area Chair (Part 2)**
>
> **4. Lack of significant ablation study and analysis.**
>
> The reviewers have requested several necessary ablation studies and analyses, which we list and address individually below.
>
> To show that although InternAgent-DR’s performance exhibits some fluctuation under network variability, it remains highly competitive, we report the **mean and variance over three runs on the GAIA benchmark**.
>
> |      | Level1         | Level2        | Level3        | Avg          |
> |------|----------------|---------------|---------------|--------------|
> | InternAgent-DR | 89.8 (±2.9)  | 78.3 (±1.8)  | 53.8 (±3.8)  | 78.2 (±0.6) |
>
> To demonstrate **the generalization capability of InternPlanner**, we also conducted tests on the GPQA datasets, and the results will be added to Table 3. As shown, InternPlanner achieves performance improvements on both GAIA and GPQA, demonstrating the strong generalization capability of its training.
>
> | Planner                 | Level1 | Level2 | Level3 | Avg   | Bio   | Chem  | Phys  | Avg.  |
> |-------------------------|--------|--------|--------|-------|-------|-------|-------|-------|
> | **Qwen3-8B**            | 58.49  | 46.51  | 11.54  | 44.85 | 52.63 | 56.99 | 75.58 | 64.65 |
> | **InternPlanner-8B (ours)** | 70.25  | 67.44  | 34.61  | 66.06 | 63.16 | 60.22 | 86.05 | 71.64 |
> | **Qwen3-32B**           | 77.36  | 67.44  | 30.77  | 64.81 | 68.42 | 59.14 | 84.88 | 71.15 |
> | **InternPlanner-32B**   | 84.91  | 70.93  | 42.31  | 70.91 | 78.95 | 67.74 | 94.19 | 80.24 |
>
> We also make a **resource comparison** in the table below. Thanks to InternAgent-DR's more cost-effective and efficient base model, we achieve advantages in both cost and response time.
>
> | Method          | API Cost ($/query) | Time (s/task) |
> |-----------------|---------------------|----------------|
> | InternAgent-DR  | 0.86                | 177            |
> | OWL             | 1.20                | 236            |
>
> These are our summarized main concerns from the reviewers and our corresponding responses. For more detailed, point-by-point explanations, we kindly invite you to refer to our individual responses to each reviewer. In addition, we would be grateful if you could also consider our note on “**Concerns Regarding AI-Generated Reviews and Review Quality in Submission #830**,” particularly regarding **the extremely low-quality reviews from Reviewer FiJA and sJkB.**

---

### Note · Program_Chairs · 2025-12-08
**Submission Desk Rejected by Program Chairs**

The submission has a link to a GitHub repo that contains author information, violating double blind. Consequently the submission must be desk rejected.